# Bypass Back-propagation: Optimization-based Structural Pruning for Large Language Models via Policy Gradient

## Abstract

In contrast to moderate-size neural network pruning, structural weight pruning on the Large-Language Models (LLMs) imposes a novel challenge on the efficiency of the pruning algorithms, due to the heavy computation/memory demands of the LLMs. Recent efficient LLM pruning methods typically operate at the post-training phase without the expensive weight finetuning, however, their pruning criteria often rely on **heuristically hand-crafted metrics**, potentially leading to suboptimal performance. We instead propose a novel **optimization-based structural pruning** that learns the pruning masks in a probabilistic space **directly by optimizing the loss of the pruned model**. To preserve the efficiency, our method **eliminates the back-propagation** through the LLM *per se* during the optimization, requiring only **the forward pass of the LLM**. We achieve this by learning an underlying `Bernoulli` distribution to sample binary pruning masks, where we decouple the `Bernoulli` parameters from the LLM loss, thus facilitating an efficient optimization via a *policy gradient estimator* without back-propagation. As a result, our method is able to 1) *operate at structural granularities of channels, heads, and layers*, 2) *support global and heterogeneous pruning* (*i.e.*, our method automatically determines different redundancy for different layers), and 3) *optionally initialize with a metric-based method* (for our `Bernoulli` distributions). Extensive experiments on LLaMA, LLaMA-2, LLaMA-3, Vicuna, and Mistral using the C4 and WikiText2 datasets demonstrate that our method operates for 2.7 hours with around 35GB memory for the 13B models on a single A100 GPU, and our pruned models outperform the state-of-the-arts *w.r.t.* both perplexity and the majority of various zero-shot tasks. Codes will be released.

## 1 Introduction

With the rapid development of Large Language Models (LLMs) Brown et al. (2020); Achiam et al. (2023) and their expanding multitude of applications across various domains, the efficiency of LLMs with vast parameters and complex architectures becomes crucial for practical deployment. In this paper, we aim to compress the LLM through structural pruning, which removes certain structural components such as channels or layers to reduce the model size with hardware-friendly acceleration.

Pioneering structural pruning in the pre-LLM era involves pruning channels or layers through *optimization*, which determines the structures to prune by back-propagating the task loss through the networks Liu et al. (2018); Blalock et al. (2020); Zhu & Gupta (2017); Louizos et al. (2017); Gale et al. (2019); Frankle & Carbin (2018). These methods operate at the in-training Huang & Wang (2018); Evci et al. (2020); Zhao et al. (2019); He et al. (2018a) or post-training Molchanov et al. (2019); Wang et al. (2021); Liu et al. (2021) phase, where the latter exhibit better efficiency without model weights update. We thus focus on post-training pruning in the following to ensure efficiency.

However, the heavy computational and memory demands of LLMs make existing *optimization-based pruning* methods (even post-training ones) less appropriate for LLM pruning in terms of efficiency. *Metric-based pruning* is introduced to alleviate this issue, which directly prunes specific network components based on carefully designed criteria, such as the importance score Sun et al. (2023); Das et al. (2023). Nonetheless, those criteria are often based on heuristics. As a re-

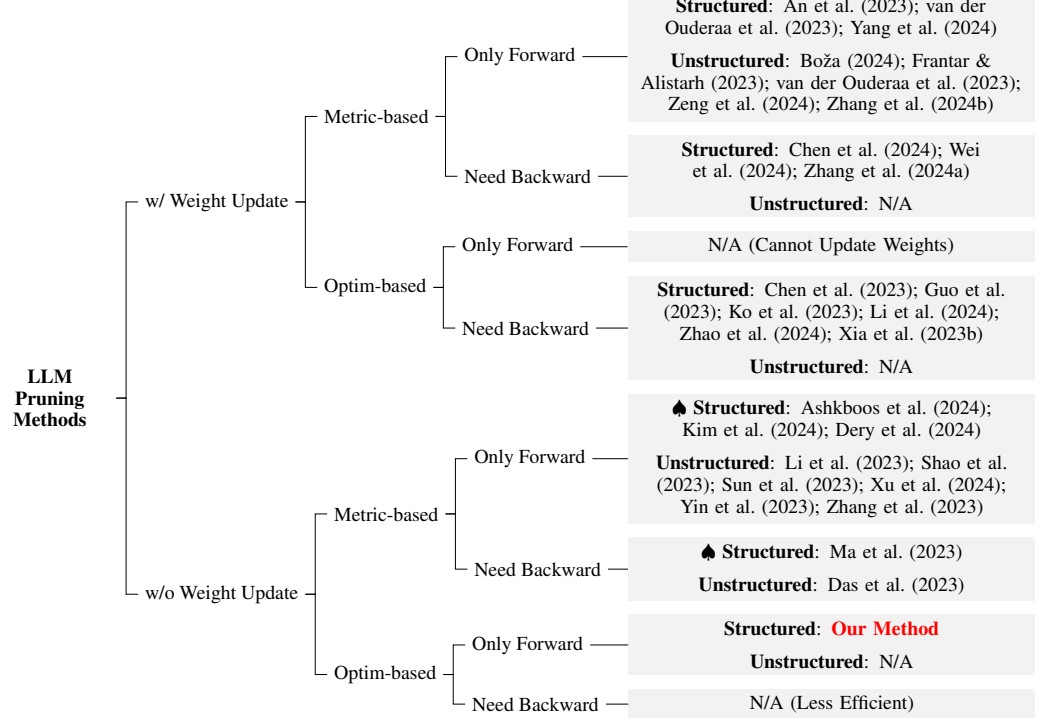

Figure 1: The taxonomy of our method among the LLM Pruning. Methods without weight update are used for comparison in our experiments (highlighted with ♠), due to the constraints on time and memory efficiency, as well as the accessibility of large-scale finetuning datasets.

sult, metric-based pruning methods often face challenges in achieving promising performance and generalizability, particularly when the pruning rate is high.

Moreover, the majority of *metric-based pruning* methods typically prune the networks by manually-designed thresholds Li et al. (2023); Zhang et al. (2023). Although different layers of LLMs may have varying levels of redundancy Yin et al. (2023); Xu et al. (2024), *achieving a global and heterogeneous pruning strategy is challenging with metric-based approaches*. This arises due to the significantly varying magnitudes of the manually designed metrics across layers, making it laborious or even impossible to set proper pruning threshold for each layer[1]. The above analysis leads to a natural question: *Can we attain the performance of **optimization-based methods** that facilities global and heterogeneous pruning without relying on hand-crafted heuristics, while preserving a similar cost with the **metrics-based methods** that is affordable on a single commercial GPU?*

In view of the above analysis, our proposed method is essentially a novel lightweight *optimization-based method*, where it 1) efficiently avoids the back-propagation through the heavy LLM, 2) naturally supports flexible pruning structural granularities such as channels, heads (of multi-head attention modules), and layers, 3) optionally can be initialized by an arbitrary metric-based approach. Particularly, our pruning efficiency is ensured using a *policy gradient estimator* Williams (1992), requiring only the LLM **forward pass** without back-propagation, which is analogous to many efficient metric-based methods and requires the same memory overhead, such as Shao et al. (2023); Men et al. (2024); Frantar & Alistarh (2023); An et al. (2023). Moreover, our method unifies the pruning of the entire LLM into a probabilistic space (that can be optionally initialized by an arbitrary metric-based approach), which eliminates the magnitude difference issue of most metric-based methods, therefore directly facilitating global and heterogeneous pruning across the entire LLM.

Specifically, we formulate our pruning as a binary mask learning/optimization problem Srinivas et al. (2017), where the binary masks determine whether to prune the corresponding structures by element-product of them. To efficiently learn those binary masks, we construct an underlying probabilistic space of `Bernoulli` distributions to sample those binary masks. By decoupling the `Bernoulli` parameters and the sampled masks, our method results in a disentanglement of the

---

[1] As a practical compromise, most metric-based methods conduct a homogeneous/uniform pruning rate for all the layers, which violates the fact that different layers could possess the different amount of redundancy.

`Bernoulli` parameters from the LLM loss, which can thus be optimized efficiently exploiting the *policy gradient estimator* in a back-propagation-free manner[2]. Moreover, the probabilistic modeling of `Bernoulli` distribution facilitates global and heterogeneous pruning across the entire LLM. Furthermore, by formulating the masks at the different structural granularities, our method supports pruning at channels, heads, and layers.

The taxonomy of our methods is illustrated in Fig. 1. In the experiments, our method is compared with SOTA structural pruning methods that *do not update the model weight simultaneously*, due to the constraints on **time and memory efficiency**[3]. We extensively validate our methods using the C4 Raffel et al. (2020) and WikiText2 Merity et al. (2016) datasets on popular LLaMA Touvron et al. (2023a), LLaMA-2 Touvron et al. (2023b), LLaMA-3 Dubey et al. (2024), Vicuna Chiang et al. (2023), and Mistral Jiang et al. (2023) models with various parameter sizes, pruning rates, and initializations, showing the promising performance and efficiency of the proposed method. For example, our method operates only 2.7 hours with about 35GB memory on a single A100 GPU to prune the LLaMA-2-13B model, resulting in clear outperformance over the SOTA methods regarding both perplexity and zero-shot performance. Our method exhibits the following features simultaneously:

- **Accuracy**, ensured by 1) our *optimization-based pruning* without heuristically hand-crafted metrics, which optionally take metric-based pruning as initialization for a better convergence, and 2) the *global and heterogeneous pruning*, as supported by our probabilistic modeling of the pruning masks.
- **Efficiency** (regarding both computations and memory), achieved by the *policy gradient estimator* for back-propagation-free and forward-only optimization *w.r.t.* the heavy LLMs.
- **Flexibility** (across various structural pruning granularities including channels, heads, and layers), coming from our *mask formulation of pruning* that can be inherently applied to different structural granularities.

## 2 RELATED WORK

Pruning has proven effective in traditional deep neural networks Han et al. (2015); Frankle & Carbin (2018); Kurtic et al. (2022); Gordon et al. (2020); Liu et al. (2019); He et al. (2018b); Zhong et al. (2018), and extensive research has been conducted on this topic. Typically, post-pruning performance is restored or even enhanced through full-parameter fine-tuning Liu et al. (2018); Blalock et al. (2020). However, for large language models (LLMs) with their vast number of parameters, full-parameter fine-tuning is computationally expensive and often impractical. To overcome this challenge, various pruning strategies Ko et al. (2023); Xia et al. (2023a); Jaiswal et al. (2023); Zhang et al. (2024a); Chen et al. (2023); Zhao et al. (2024); Ashkboos et al. (2024); Dery et al. (2024); Men et al. (2024); Fang et al. (2024); Sreenivas et al. (2024); Shen et al. (2024); Song et al. (2024); Gromov et al. (2024); Liu et al. (2024) have been developed for LLMs in recent years. These strategies can be categorized into metric-based pruning and optimization-based pruning.

**Metric-based Pruning.** Metric-based pruning methods focus on designing importance metrics for model weights or modules. The most representative method is Wanda Sun et al. (2023). It introduces a simple but effective pruning metric by considering both the magnitude of weights and activations without updating model parameters. LLM-Pruner Ma et al. (2023) efficiently trims LLMs by pinpointing and eliminating non-essential coupled structures, assessing weight importance via loss change, and using Taylor expansion to preserve performance. SparseGPT Frantar & Alistarh (2023) propose an efficient technique for estimating the Hessian matrix to reconstruct the model. These methods use pre-defined pruning metrics and often face challenges with high pruning rates. Most recently, Dery et al. (2024) proposes a structured pruning method with only forward passes with promising performance. However, it still relies on regressing the heuristically hand-crafted criteria, *e.g.*, the utility of the pruned sub-networks. Moreover, Dery et al. (2024) also leverages additional assumptions that might not hold universally, *e.g.*, the network's utility as a linear summation of building elements' utilities, and the building element's utility remaining the same/average-able across different sub-networks.

---

[2]We note that our formulation can also be interpreted from a reinforcement learning (with dense rewards) perspective in terms of Markov Decision Process (MDP), please refer to Appendix A.3 for details.

[3]After pruning is performed, it becomes affordable to finetune the weights of the pruned smaller model on a single commercial GPU, we include this "final" performance with pruning then finetuning in our experiments.

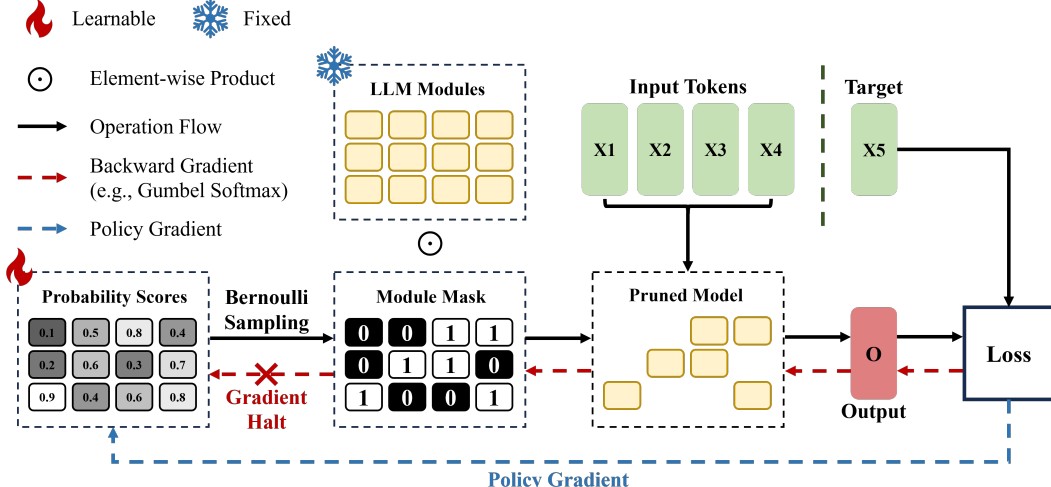

Figure 2: The overview of our method. We formulate LLM pruning as optimizing underlying `Bernoulli` distributions that sample binary masks. Being different from the conventional back-propagation method (*e.g.*, through *Gumbel Softmax* as shown by the red-dashed-arrows), our formulation decouples the masks and the `Bernoulli` parameters from the LLM loss (see Eq. (4) and Remark 3), facilitating efficient and unbiased *policy gradient* (the blue-dashed-arrow) without back-propagation through the LLM (see Eq. (5) and Remark 4).

**Optimization-based Pruning.** Optimization-based pruning methods focus on determining the model mask in an optimized manner and also involve updating the model parameters. Sheared LLaMA Xia et al. (2023b) learns pruning masks to find a subnetwork that fits a target architecture and maximizes performance, utilizing dynamic batch loading for efficient data usage. Compresso Guo et al. (2023) optimizes LLM pruning by integrating LoRA Hu et al. (2022) with L0 regularization and dynamically updates parameters during instruction tuning to enhance post-pruning performance and adaptability. LoRAShear Chen et al. (2023) and APT Zhao et al. (2024) also utilize LoRA in the pruning process alongside weight updating. However, these methods, including the optimization process and weight finetuning, invariably rely on back-propagation, which is time and memory-intensive. We propose using policy gradient estimation in the optimization process as an alternative to back-propagation, which significantly reduces the computational demands.

## 3 METHODOLOGY

We introduce our optimization-based pruning for LLMs, which is efficient without back-propagation through the LLM. We detail our probabilistic mask modeling in Sect. 3.1, and our optimization via policy gradient estimator in Sect. 3.2. The overview of our method is illustrated in Fig. 2.

### 3.1 PRUNING VIA PROBABILISTIC MASK MODELING

We formulate the network pruning as seeking binary masks Srinivas et al. (2017) to determine whether the corresponding structure should be pruned or retrained. Those binary masks are further modeled by/sampled from underlying `Bernoulli` distributions stochastically. Such formulation possesses several merits: 1) the mask formulation enables flexible pruning at channels, heads (of Multi-Head Attention, MHA), and layers; 2) the probabilistic `Bernoulli` modeling facilitates global and heterogeneous pruning across the entire LLM; and 3) our stochastical sampling decouples `Bernoulli` parameters and the sampled masks from LLM loss thus empowers an efficient *policy gradient* optimization without back-propagate through the LLM (see Sect. 3.2).

Specifically, denoting the calibration dataset with $N$ *i.i.d.* samples as $\mathcal{D} = \{(\mathbf{x}_i, \mathbf{y}_i)\}_{i=1}^N$, $\mathbf{w} = \{\mathbf{w}_i\}_{i=1}^n$ as the complete and non-overlapped modules of a LLM with model size $n$, and $\mathbf{m} = \{\mathbf{m}_i\}_{i=1}^n \in \{0,1\}^n$ as the corresponding binary masks, where $\mathbf{m}_i = 0$ implies $\mathbf{w}_i$ is pruned and otherwise $\mathbf{w}_i$ is retained. We note that $\mathbf{w}_i$ and $\mathbf{m}_i$ can be defined with various granularities such as channels, heads, and layers for structural pruning[4]. Then, our structural pruning of LLMs can be

---

[4]Note that for the channel and head granularities, we prune the dimensions of the hidden states following Ma et al. (2023); An et al. (2023), without altering the final output channels for each block and thus ensuring the feasibility of the residue connections, as shown in the conceptual figure in Appendix A.2.

formulated as a binary optimization with constraints:

$$\min_{\mathbf{m}} \ \mathcal{L}(\mathcal{D}; \mathbf{w} \odot \mathbf{m}) := \frac{1}{N} \sum_{i=1}^{N} \ell(f(\mathbf{x}_i; \mathbf{w} \odot \mathbf{m}), \mathbf{y}_i), \tag{1}$$

$$\text{s.t. } \|\mathbf{m}\|_1 \leq rn \ \text{ and } \ \mathbf{m} \in \{0, 1\}^n.$$

where $f(\cdot; \mathbf{w} \odot \mathbf{m})$ is the pruned network, $\ell(\cdot, \cdot)$ is the loss function, *e.g.*, the cross-entropy loss, and $r$ is the target pruning rate. We note that the binary optimization problem Eq. (1), *i.e.*, finding high-quality masks $\mathbf{m}$ from the discrete and exponentially growing solution space, is typically NP-hard.

Therefore, we relax the discrete optimization using a probabilistic approach, by treating $n$ masks as binary *random variables* sampled from $n$ underlying `Bernoulli` distributions with parameters $\mathbf{s} = \{\mathbf{s}_i\}_{i=1}^n \in [0, 1]^n$. This yields the conditional distribution of $\mathbf{m}$ over $\mathbf{s}$:

$$p(\mathbf{m}|\mathbf{s}) = \prod_{i=1}^{n} (s_i)^{m_i}(1 - s_i)^{1-m_i}. \tag{2}$$

By relaxing the $\ell_1$ norm in Eq. (1) using its expectation, *i.e.*, $\|\mathbf{m}\|_1 \approx \mathbb{E}_{\mathbf{m} \sim p(\mathbf{m}|\mathbf{s})} \|\mathbf{m}\|_1 = \sum_{i=1}^{n} s_i = \mathbf{1}^\top \mathbf{s}$, we have the following excepted loss minimization problem:

$$\min_{\mathbf{s}} \ \mathbb{E}_{p(\mathbf{m}|\mathbf{s})} \mathcal{L}(\mathcal{D}; \mathbf{w} \odot \mathbf{m}), \tag{3}$$

$$\text{s.t. } \mathbf{1}^\top \mathbf{s} \leq rn \ \text{ and } \ \mathbf{s} \in [0, 1]^n.$$

**Remark 1** *Problem* (3) *is a continuous relaxation of the discrete Problem* (1). *The feasible region of Problem* (3) *is as simple as the intersection of the cube* $[0, 1]^n$ *and the half-space* $\mathbf{1}^\top \mathbf{s} \leq rn$. *Moreover, the parameterization of Problem* (3) *in the probabilistic space facilitates automatically learning the redundancy across different layers for global and heterogeneous pruning.*

### 3.2 OPTIMIZATION BY POLICY GRADIENT ESTIMATOR

Conventional neural network training paradigm usually adopts back-propagation to estimate the gradient of Eq. (3), *e.g.*, through Gumbel-Softmax Maddison et al. (2016); Dupont et al. (2022) which reparameterizes the mask $\mathbf{m}$ as a function of $\mathbf{s}$, *i.e.*, $m_i = \phi(s_i)$ or $m_i = \phi(s_i, \epsilon)$ with $\epsilon \sim \mathcal{N}(0, 1)$. However, the back-propagation has the following intrinsic issues in LLM pruning.

**Remark 2** *Intrinsic issues of back-propagation in LLM pruning: 1) the back-propagation is computationally expensive and costs a large amount of memory; 2) the computation of gradients can not be satisfied by using the sparsity in $\mathbf{m}$, i.e., $\frac{\partial m_i}{\partial s_i} \neq 0$ even if $m_i = 0$. In other words, one has to go through the full model for back-propagation even when lots of the LLM modules have been masked.*

Now we present our efficient (back-propagation-free) and unbiased optimization for Problem (3). We proposed to adopt Policy Gradient Estimator (PGE) to estimate the gradient with only forward propagation, avoiding the pathology of the chain-rule-based estimator. Specifically, in order to update the `Bernoulli` parameters $\mathbf{s}$, we have the following objective $\Phi(\mathbf{s})$:

$$\Phi(\mathbf{s}) = \mathbb{E}_{p(\mathbf{m}|\mathbf{s})} \mathcal{L}(\mathcal{D}; \mathbf{w} \odot \mathbf{m}) = \int p(\mathbf{m}|\mathbf{s}) \mathcal{L}(\mathcal{D}; \mathbf{w} \odot \mathbf{m}) d\mathbf{m}, \tag{4}$$

$$\text{s.t. } \mathbf{1}^\top \mathbf{s} \leq rn \ \text{ and } \ \mathbf{s} \in [0, 1]^n.$$

Our key idea is that in Eq. (4), the score vector $\mathbf{s}$ only appears in the conditional probability $p(\mathbf{m}|\mathbf{s})$ for sampling $\mathbf{m}$, which is decoupled from the network loss term $\mathcal{L}(\mathcal{D}; \mathbf{w} \odot \mathbf{m})$.

**Remark 3** *Differences with Gumbel Softmax: 1) As shown in Eq. (4), our PGE formulates the mask $\mathbf{m}$ as a random variable which is only related to the distribution $\mathbf{s}$ through the conditional probability $p(\mathbf{m}|\mathbf{s})$ of probabilistic sampling. Therefore, the expensive back-propagation through the LLM can be omitted in gradient estimation using the PGE. On the contrary, in the Gumbel-Softmax estimator, $\mathbf{m}$ is a function of $\mathbf{s}$, requiring the back-propagation through the whole networks (see the blue and red gradient flows in Fig. 2). 2) As a result, Gumbel-Softmax is challenged by the back-propagation issues discussed in Remark 2. 3) Gumbel-Softmax is known to be biased especially when the temperature is high Huijben et al. (2022). 4) The vanilla PGE might suffer from large variance Liu et al. (2020), we thus exploit a variance-reduced PGE discussed later in Eq. (7) with theoretical analysis and empirical ablations in Appendices A.4 and A.10.*

Specifically, the optimization of Eq. (4) via the policy gradient estimator holds that:

$$\nabla_{\mathbf{s}}\Phi(\mathbf{s}) = \int \mathcal{L}(\mathbf{m})\nabla_{\mathbf{s}}p(\mathbf{m}|\mathbf{s}) + \underbrace{p(\mathbf{m}|\mathbf{s})\nabla_{\mathbf{s}}\mathcal{L}(\mathbf{m})}_{=\,0}\,\mathrm{d}\mathbf{m}$$

$$= \int \mathcal{L}(\mathbf{m})p(\mathbf{m}|\mathbf{s})\nabla_{\mathbf{s}}\log(p(\mathbf{m}|\mathbf{s}))\mathrm{d}\mathbf{m} \qquad (5)$$

$$= \mathbb{E}_{p(\mathbf{m}|\mathbf{s})}\mathcal{L}(\mathbf{m})\nabla_{\mathbf{s}}\log(p(\mathbf{m}|\mathbf{s})).$$

The last equality shows that $\mathcal{L}(\mathbf{m})\nabla_{\mathbf{s}}\log(p(\mathbf{m}|\mathbf{s}))$ is an unbiased stochastic gradient for $\Phi(\mathbf{s})$.

**Remark 4** *The efficiency of Eq.* (5): *1) Equation* (5) *can be computed purely with forward propagation. 2) The computation cost for the gradients,* i.e., $\nabla_{\mathbf{s}}\log(p(\mathbf{m}|\mathbf{s})) = \frac{\mathbf{m}-\mathbf{s}}{\mathbf{s}(1-\mathbf{s})}$, *is negligible. Therefore, our PGE is much efficient compared to the backward-propagation-based estimators.*

The stochastic gradient descent algorithm in the batch-training paradigm is:

$$\mathbf{s} \leftarrow \mathbf{proj}_{\mathcal{C}}(\mathbf{z}) \text{ with } \mathbf{z} := \mathbf{s} - \eta\mathcal{L}(\mathcal{D}_B; \mathbf{w} \odot \mathbf{m})\nabla_{\mathbf{s}}\log(p(\mathbf{m}|\mathbf{s})). \qquad (6)$$

where $\mathcal{D}_B = \{(\mathbf{x}_i, \mathbf{y}_i)\}_{i=1}^B$ is batch samples from $\mathcal{D}$ with batch size $B$, and $\mathcal{L}(\mathcal{D}_B; \mathbf{w}\odot\mathbf{m})$ is the loss on $\mathcal{D}_B$ with the pruned model by masks $\mathbf{m}$. The projection operator $\mathbf{proj}_{\mathcal{C}}(\cdot)$ is to ensure the updated scores $\mathbf{s}$ to be constrained in the feasible domain $\mathcal{C}$ that satisfies $\mathcal{C} = \{\mathbf{1}^\top\mathbf{s} \leq K\}\bigcap\{\mathbf{s} \in [0,1]^n\}$. We implement the projection operator from Wang & Carreira-Perpinán (2013), the details of which can be found in Appendix A.1.

Policy gradient might suffer from large variance Liu et al. (2020). To reduce the variance for fast and stable training, we minus a moving average baseline Zhao et al. (2011) which is calculated by 1) obtaining the averaged loss of multiple sampling trials, then 2) taking the moving average of the current and the previous losses given a window size. Denote the baseline as $\delta$, given window size $T$ (set to 5), and mask sampling times $N_s$ (set to 2), we update $\mathbf{s}$ in each training step via Eqs. (7) and (8). The theoretical analysis and empirical ablations can be found in Appendix A.4 and A.10.

$$\mathbf{s} \leftarrow \mathbf{proj}_{\mathcal{C}}(\mathbf{z}) \text{ with } \mathbf{z} := \mathbf{s} - \eta[\frac{1}{N_s}\sum_{i=1}^{N_s}(\mathcal{L}(\mathcal{D}_B; \mathbf{w} \odot \mathbf{m}^{(i)})-\delta)\nabla_{\mathbf{s}}\log(p(\mathbf{m}^{(i)}|\mathbf{s}))]. \qquad (7)$$

$$\delta \leftarrow \frac{T-1}{T}\delta + \frac{1}{N_sT}\sum_{i=1}^{N_s}\mathcal{L}(\mathcal{D}_B; \mathbf{w} \odot \mathbf{m}^{(i)}). \qquad (8)$$

Our efficient pruning algorithm is summarized in Appendix A.1. Note that our formulation can also be interpreted as a dense rewards reinforcement learning problem, as discussed in Appendix A.3.

**Initialization.** Algorithms based on policy gradient usually require an effective initialization to get enhanced results. In this context, previous hand-crafted pruning metric can be applied to initialize the probability of each module: $\mathbf{s}_0 \leftarrow \sigma(\mathbf{x})$, in which $\mathbf{x}$ can be any pruning metric derived from existing method, $\mathbf{s}_0$ represents the initial probability assigned to each module, and $\sigma$ symbolizes a non-linear transformation. **We note that initializing from a prior metric-based method is only an option, while a random initialization strategy can already produce good performance.** Please refer to different initializations $\mathbf{x}$ and transformations $\sigma$ discussed in Appendices A.12 and A.11.

**Applicability of PGE in Learning Pruning Masks.** We note that the precision of PGE may not match that of conventional back-propagation. Given that we are learning the **binary** masks $\mathbf{m}$ (distinct from the **float** weights), it is expected that the precision requirement of $\mathbf{s}$ can be modest. Moreover, our PGE is unbiased (compared to the biased Gumbel Softmax). These factors make the PGE suitable for learning the masks, which is empirically validated with extensive experiments.

## 4 EXPERIMENTS

Extensive experiments have been conducted in this section to validate the promising performance of the proposed method. In short, our method has been validated across **different LLM models with various sizes**, **pruning rates**, **structural granularities for pruning** (*i.e.*, channels, heads, and layers), and **initializations** (in the ablation analysis). In the following, we first detail our experimental

setups in Sect. 4.1. After that, our main results against the state-of-the-art methods for channels and heads pruning, as well as layers pruning, are shown in Sects. 4.2 and 4.3. We illustrate the zero-shot performance in Sect. 4.4, Appendices A.6 and A.7. Our method runs 2.7 hours for LLaMA-2-13B with a similar GPU memory (*i.e.*, ∼35GB) as Wanda-sp An et al. (2023) as shown in Appendix A.5. Considering the constraints on computations and memory, we compare with the state-of-the-art methods without in-pruning weight updates, and we report the *pruning then finetuning* performance in Appendix A.6, as it becomes affordable to finetune a smaller model after pruning. We also show multiple-run statistics of our method in Appendix A.9.

## 4.1 EXPERIMENTAL SETUPS

**Structural Granularities for Pruning.** We validate our method on various structural granularities for pruning, namely *channels, heads*, and *layers*. For the effects of different initializations, we extensively investigated them in Sect. 5 and Appendices A.11 and A.12.

*Head and Channel Granularities*: We follow Ma et al. (2023); Zhang et al. (2024a) to prune the *heads* of the multi-head attention (MHA) modules and the *channels* of the MLP modules in Sect. 4.2. We initialize our methods with an efficient metric-based structural pruning method, *i.e.*, Wanda-sp An et al. (2023). Our method is compared to the state-of-the-art Wanda-sp An et al. (2023), LLM-Pruner Ma et al. (2023), SliceGPT Frantar & Alistarh (2023), and Bosai Dery et al. (2024).

*Layer Granularity*: We also validate the layer granularity by pruning the entire transformer layer consisting of an MHA module and an MLP module. Note that pruning on the structural granularity of layers is less exploited for LLMs, thus in this experiment, we use the lightweight Layerwise-PPL Kim et al. (2024) for initialization, and compare our method with Layerwise-PPL Kim et al. (2024).

**LLM Models and Sizes.** LLaMA-{7B, 13B} Touvron et al. (2023a), LLaMA-2-{7B, 13B} Touvron et al. (2023b), LLaMA-3-8B Dubey et al. (2024), Vicuna-{7B, 13B} Chiang et al. (2023), and Mistral-7B-Instruct-v0.3 Jiang et al. (2023) are used as the source models in our experiments.

**Pruning Rate.** Promising performance with a high pruning rate could be challenging to obtain when employing metric-based pruning, owing to the heuristically designed metrics. To validate the superior performance of our optimization-based pruning under this situation, we select high pruning rates ranging from 30% to 50%, *i.e.*, **structurally** removing 30% to 50% model parameters.

**Datasets.** We perform the experiments following the cross-dataset settings in Sun et al. (2023), where the C4 dataset Raffel et al. (2020) is used for training and the WikiText2 dataset Merity et al. (2016) is used for evaluation. This challenging cross-dataset setup potentially better reflects the generalization of the pruned model.

**Training and Evaluation Details.** We update the underlying `Bernoulli` distributions (for mask sampling) simply using SGD with a learning rate of 6e-3 for LLaMA-3 experiments and 2e-3 for the remaining. The batch size is fixed to 8 and we train our lightweight policy gradient estimator for 1 epoch on the C4 dataset with 120K segments, in which each segment has a sequence length of 128.

To reduce the evaluation variance, we deterministically construct the pruned architecture for evaluation, *i.e.*, given a prune rate $r$, we first rank all the $\mathbf{s}$, then deterministically set $\mathbf{m}$ corresponding to the minimal $r\%$ of $\mathbf{s}$ as 0 (otherwise 1). We report the perplexity on the WikiText2 dataset using a sequence length of 128. Given a tokenized sequence $\mathbf{X} = (x_0, x_1, \ldots, x_t)$, the perplexity of $\mathbf{X}$ is:

$$\text{Perplexity}(\mathbf{X}) = \exp\left\{-\frac{1}{t}\sum_{i}^{t}\log p_\theta(x_i|x_{<i})\right\},$$

where $\log p_\theta(x_i|x_{<i})$ is the log-likelihood of token $x_i$ conditioned on the preceding tokens $x_{<i}$.

## 4.2 RESULTS ON CHANNELS AND HEADS PRUNING

The results of channels and heads pruning following Ma et al. (2023); Zhang et al. (2024a) are shown in Fig. 3. Our method achieves the lowest perplexity scores. It verifies the superiority of optimization-based global and heterogeneous pruning. Especially, such outperformance is more significant at larger pruning rates over 40%. The results on Mistral-7B-Instruct-v0.3 Jiang et al. (2023) are shown in Table A6 of Appendix A.8.

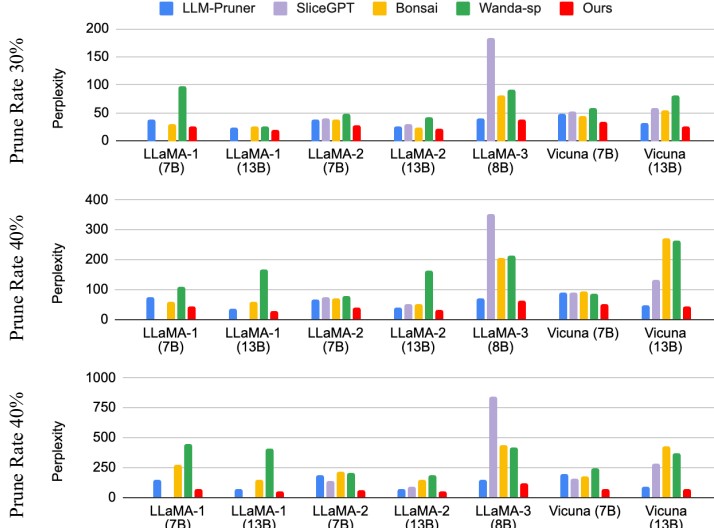

Figure 3: Results on *channels and heads* pruning. Our method is initialized by Wanda-sp (please also refer to Sect. 5.1 and Appendix A.12 for a detailed discussion about initializations). All the methods are calibrated using the C4 dataset and validated on the WikiText2 dataset *w.r.t.* perplexity.

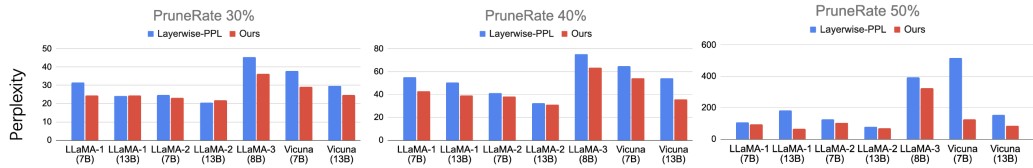

Figure 4: Results on *layers* pruning. Our method is initialized by Layerwise-PPL (please also refer to Sect. 5.1 and Appendix A.12 for detailed discussion about initializations). All the methods are calibrated using the C4 dataset and validated on the WikiText2 dataset *w.r.t.* perplexity.

### 4.3 RESULTS ON LAYER PRUNING

We illustrate the results on layer pruning in Fig. 4, which demonstrate that our method significantly outperforms the baseline method with pruning rates larger than 40%. For LLaMA-13B and LLaMA-2-13B with moderate pruning rates of 30%, our method works comparable with Layerwise-PPL, which might imply that the searching space is small with the coarse layer granularity, and the larger 13B models with more redundancy works well with metric-based pruning with smaller pruning rate.

### 4.4 ZERO-SHOT PERFORMANCE ON OTHER NLP TASKS

We follow SliceGPT Ashkboos et al. (2024) to assess our pruned LLM using EleutherAI LM Harness Gao et al. (2023) on five zero-shot tasks: PIQA Bisk et al. (2020), WinoGrande Sakaguchi et al. (2021), HellaSwag Zellers et al. (2019), ARC-e and ARC-c Clark et al. (2018). We also report the average scores across the five tasks. Our results on LLaMA-3-8B and LLaMA-2-7B are shown in Tables 1 and A5 of Appendix A.7, which demonstrate that our overall performance is superior to the baselines, though using only the C4 dataset for pruning might introduce a negative influence on some particular cross-dataset zero-shot tasks such as Hellaswag Zellers et al. (2019).

## 5 ABLATION ANALYSIS

For ablations, we investigate 1) effect of various initialization of our method in Sect. 5.1 and Appendices A.11 and A.12, 2) performance of global and heterogeneous pruning versus that of local and homogenous pruning in Sect. 5.2, 3) analysis of the remaining modules after pruning in Sect. 5.3 and Appendix A.13, and 4) effect of the variance-reduced policy gradient in Appendix A.10.

### 5.1 DIFFERENT INITIALIZATIONS

Our `Bernoulli` policy requires initialization to perform policy gradient optimization and to sample pruning masks. In this section, we investigate the **effect** and the **necessity** of using different metric-based methods as initializations. Moreover, the initialization of the `Bernoulli` policy

Table 1: Perplexity (PPL) and zero-shot accuracies (%) of LLaMA-3-8B for 5 tasks.

| Method | PruneRate | PPL ↓ | PIQA | HellaSwag | WinoGrande | ARC-e | ARC-c | Average |
|--------|-----------|-------|------|-----------|------------|-------|-------|---------|
| Dense | 0% | 14.13 | 79.71 | 60.19 | 72.61 | 80.09 | 50.34 | 68.59 |
| LLM-Pruner | | 40.18 | 71.38 | 37.84 | 55.64 | 57.78 | 27.21 | 49.97 |
| SliceGPT | | 183.94 | 68.34 | **53.92** | 57.57 | 49.41 | 28.07 | 51.39 |
| Bonsai | 30% | 80.89 | 64.53 | 36.10 | 55.09 | 47.64 | 22.52 | 45.18 |
| Wanda-sp | | 92.14 | 59.74 | 31.46 | 52.64 | 44.02 | 19.88 | 41.55 |
| Ours | | **38.99** | **72.25** | 43.56 | **59.04** | **59.85** | **29.44** | **52.83** |
| LLM-Pruner | | 70.60 | 66.26 | 31.90 | 54.06 | 49.74 | 22.52 | 44.90 |
| SliceGPT | | 353.09 | 61.53 | **39.98** | 52.80 | 36.66 | **25.17** | 43.23 |
| Bonsai | 40% | 204.61 | 58.81 | 29.43 | 48.93 | 33.21 | 18.15 | 37.71 |
| Wanda-sp | | 213.47 | 56.58 | 27.46 | 50.35 | 32.07 | 17.06 | 36.70 |
| Ours | | **63.85** | **67.63** | 37.36 | **56.91** | **50.67** | 24.91 | **47.50** |
| LLM-Pruner | | 145.65 | 61.15 | 29.10 | **51.93** | 39.98 | 19.36 | 40.30 |
| SliceGPT | | 841.20 | 56.37 | **32.66** | 48.38 | 32.45 | **22.10** | 38.39 |
| Bonsai | 50% | 440.86 | 55.66 | 26.94 | 50.51 | 30.64 | 17.83 | 36.32 |
| Wanda-sp | | 413.86 | 55.39 | 27.07 | 49.72 | 29.59 | 18.26 | 36.01 |
| Ours | | **119.75** | **62.51** | 30.89 | 51.85 | **41.12** | 20.65 | **41.40** |

Table 2: Channels and heads pruning results with *different initializations* on LLaMA-2-7B. **Bold** and Underscored denote the first and second best results, respectively.

| Method | PruneRate | Perplexity | PruneRate | Perplexity | PruneRate | Perplexity |
|--------|-----------|------------|-----------|------------|-----------|------------|
| LLM-Pruner | | 38.94 | | 68.48 | | 190.56 |
| SliceGPT | 30% | 40.40 | 40% | 73.76 | 50% | 136.33 |
| Bonsai | | 39.01 | | 69.18 | | 216.85 |
| Wanda-sp | | 49.13 | | 78.45 | | 206.94 |
| Ours (Random Init) | 30% | 37.24 | 40% | 60.16 | 50% | 160.75 |
| Ours (Random-Prog. Init) | | 31.43 | | 49.86 | | 86.55 |
| Ours (LLM-Pruner Init) | 30% | 35.75 | 40% | 65.32 | 50% | 116.80 |
| Ours (Wanda-sp Init) | | **28.18** | | **39.81** | | **65.21** |

should be probabilistic values between 0 and 1, but the metrics calculated by the metric-based methods Sun et al. (2023); An et al. (2023); Ma et al. (2023) may not hold this range. We thus discuss **different projection strategies** that transform those metrics to [0, 1] in Appendix A.11.

In order to deal with the practical case when a metric-based pruning is not *apriori*, we propose progressive pruning with random initialization (*Random-Progressive*), which is trained progressively with increasing pruning rates (each for only $1/3$ epoch). Details can be found in Appendix A.12.

Different initializations are tested on LLaMA-2-7B. The baselines include the simple random initialization with the target pruning rate (*Random*) and the progressive pruning with random initialization (*Random-Progressive*). For the pruning granularity of channels and heads, we investigate the initializations from Wanda-sp An et al. (2023) and LLM-Pruner[5] Ma et al. (2023), as shown in Table 2. While for the layers granularity, we test the initialization using Layerwise-PPL Kim et al. (2024), as shown in Table A11 of Appendix A.12.

Our results in Tables 2 and A11 demonstrate that 1) different initializations lead to different results, 2) compared to the state-of-the-art methods, our method with most initializations except the random one exhibit new state-of-the-art results, and 3) The proposed *Random-Progressive* initialization ranks the second place in most cases, surpassing previous state-of-the-art methods, **which suggests less necessity for employing a prior metric-based method to initialize our algorithm**.

## 5.2 MERITS OF GLOBAL AND HETEROGENEOUS PRUNING

Our method is able to perform global and heterogeneous pruning throughout the entire network, which can be difficult for metric-based pruning methods Sun et al. (2023); Frantar & Alistarh (2023); Shi et al. (2023), as the calculated metrics across different layers oftentimes exhibit different magnitudes. As a compromise, those metric-based methods prune each layer locally and homogeneously.

We validate the merits of global and heterogeneous pruning over local and homogeneous pruning, where we compare our method with a variant in which we prune each layer homogeneously. The

---

[5]We follow LLM-Pruner Ma et al. (2023) to fix the first four and the last two layers from pruning.

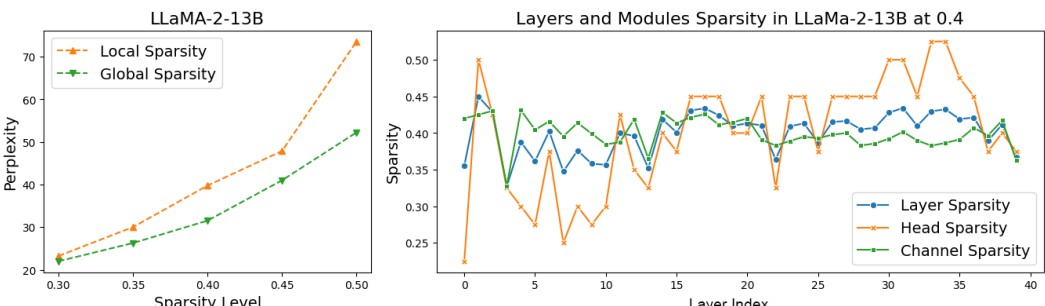

Figure 5: Global vs. local pruning.    Figure 6: Channels, heads, and layers sparsities.

channels and heads pruning results on LLaMA-2-13B are shown in Fig. 5, demonstrating that the global and heterogeneous pruning significantly outperforms its local and homogeneous counterpart.

### 5.3 ANALYSIS OF THE POST-PRUNING MODULES

As global and heterogeneous pruning is performed through our optimization, it is interesting to investigate the pruned modules in each layer. We show the channel, heads, and layers sparsity (*i.e.*, the pruned portion of the corresponding granularity) on LLaMA-2-13B with channels and heads pruning at 40% in Fig. 6. Those results on LLaMA-2-7B are shown in Fig. A2 of Appendix A.13.

Figures 6 and A2 demonstrate that the pruned LLM exhibits low sparsity in the first and last layers, which is consistent with the previous studies that these layers have a profound impact on the performance of LLMs Ma et al. (2023). Moreover, it can be observed that the heads (of MHA) granularity exhibits lower sparsity in the shallow layers (especially in the first layer), while such observation does not hold for the channels (of MLP) granularity. In other words, the pruned sparsity of the channel granularity is more evenly distributed whereas the deeper layers have a slightly less sparsity. This might imply that the shallow layers focus more on attention, while the deeper layer imposes slightly more responsibility for lifting the feature dimensions through MLP.

### 6 DISCUSSION AND CONCLUSION

**Limitations and Future Works.** Firstly, as an optimization-based pruning, our method *requires more (but affordable) training time* for optimization (*e.g.*, 2.7 hours for LLaMA-2-13B) compared to the (heuristic) metric-based methods, while our method benefits from *significantly improved performance* with a *similar memory complexity* (35GB for LLaMA-2-13B, as both only require forward).

Secondly, there exist advanced policy gradient algorithms with potentially lower variance from the reinforcement learning community Schulman et al. (2017). As 1) the primary focus of this paper is on the back-propagation-free formulation of the LLM pruning problem, and 2) our formulation guarantees dense rewards at each step, we thus use a basic policy gradient algorithm similar to REINFORCE Williams (1992) with simple variance reduction using a moving average baseline. We leave exploiting more powerful policy gradient algorithms as our future work.

Lastly, the performance of the proposed method on specific domains/tasks can rely heavily on the availability of domain-specific datasets. Though the cross-dataset evaluation is verified *w.r.t.* perplexity, using only the C4 dataset for pruning might introduce a negative influence on some particular cross-dataset zero-shot tasks such as WinoGrande Sakaguchi et al. (2021) and Hellaswag Zellers et al. (2019), as shown in the Tables 1 and A5 of Appendix A.7.

**Conclusion.** We propose an efficient optimization-based structural pruning method for LLMs, which 1) does not need back-propagation through the LLM *per se*, 2) enables global and heterogeneous pruning, and 3) supports pruning granularities of channels, heads, and layers. Our method can optionally take a metric-based pruning as initialization to achieve a further improved performance. We implement our method by learning an underlying `Bernoulli` distribution of binary pruning mask. As we decouple the `Bernoulli` parameter and the sampled masks from the LLM loss, the `Bernoulli` distribution can thus be optimized by a policy gradient estimator without back-propagation through the LLM. Our method operates for 2.7 hours with 35GB of memory on a single A100 GPU to prune the LLaMA-2 13B model. Extensive experiments on various LLM models and sizes with detailed ablation analysis validate our promising performance.

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

# A APPENDIX

We discuss the following additional analysis, results, and ablations in the appendices.

**Analysis**:

1. Projection operator for sparsity constraint and the overall algorithm in Appendix A.1.

2. Details on hidden states pruning for channel and head granularities in Appendix A.2.

3. A reinforcement learning perspective of the proposed method in Appendix A.3.

4. Theoretical analysis of moving average baseline for policy gradient in Appendix A.4.

**Results**:

1. Statistics of the training time & memory, and the inference latency in Appendix A.5.

2. Performance after pruning and (then) finetuning in Appendix A.6.

3. Zero-shot performance on LLaMA-2-7B in Appendix A.7.

4. Pruning performance on Mstral-7B-Instruct-V0.3 in Appendix A.8.

5. Random error-bar statistics in Appendix A.9.

6. Hamming distance of the masks generated from different methods in Appendix A.14.

7. Performance using the same amount of calibration data in A.15.

**Ablations**:

1. Ablations on the moving average baseline for policy gradient in Appendix A.10.

2. Ablations on projection strategy for initialization: from metric to probability in Appendix A.11.

3. More ablations with different initializations in Appendix A.12.

4. More ablations of the post-pruning modules on LLaMA-2-7B in Appendix A.13.

## A.1 PROJECTION OPERATOR FOR SPARSITY CONSTRAINT AND THE OVERALL ALGORITHM

**Details of the Projection Operator.** In our proposed probabilistic framework, the sparsity constraint manifests itself in a feasible domain on the probability space defined in Problem (3). We denote the feasible domain as $\mathcal{C} = \left\{ \mathbf{1}^\top \mathbf{s} \leq K \right\} \bigcap \left\{ \mathbf{s} \in [0,1]^n \right\}$. The theorem Wang & Carreira-Perpinán (2013) below shows that the projection of a vector onto $C$ can be calculated efficiently.

**Theorem 1.** *For each vector $\mathbf{z}$, its projection $\boldsymbol{proj}_{\mathcal{C}}(\mathbf{z})$ in the set $C$ can be calculated as follows:*

$$\mathbf{proj}_{\mathcal{C}}(\mathbf{z}) = \min(1, \max(0, \mathbf{z} - v_2^* \mathbf{1})) \tag{A1}$$

*where $v_2^* = max(0, v_1^*)$ with $v_1^*$ being the solution of the following equation*

$$\mathbf{1}^T \left[ \min(1, \max(0, \mathbf{z} - v_1^* \mathbf{1})) \right] - K = 0 \tag{A2}$$

Equation (A1) can be solved by the bisection method efficiently.

The theorem above as well as its proof is standard and it is a special case of the problem stated in Wang & Carreira-Perpinán (2013). This component, though not the highlight of our work, is included for the reader's convenience and completeness.

**Algorithm.** The pseudo-code of our overall algorithm is detailed below.

---

**Algorithm 1** Pseudo-code of PG pruning

---

**Input:** target remaining ratio $r > 0$, a dense pretrained network $\mathbf{w}$, the step size $\eta > 0$, mini-batch size $B > 0$, moving average window size $T$, and calibration dataset $\mathcal{D}$
**Initialize:** Init probability $\mathbf{s}$ from any pruning metric $\mathbf{x}$, ans set moving average $\delta = 0$
1: **while** until convergence **do**
2:  Sample a mini-batch from the entire calibration dataset: $\mathcal{D}_B = \{(\mathbf{x}_i, \mathbf{y}_i)\}_{i=1}^{B} \sim \mathcal{D}$
3:  Sample $\mathbf{m}^{(i)}$ from $p(\mathbf{m}|s)$, $i = 1, 2, \ldots, N_s$
4:  Update the moving average baseline $\delta$ via Eq. (8)
5:  Uptate $\mathbf{s}$ via Eqs. (7), (A1), and (A2).
6: **end while**

---

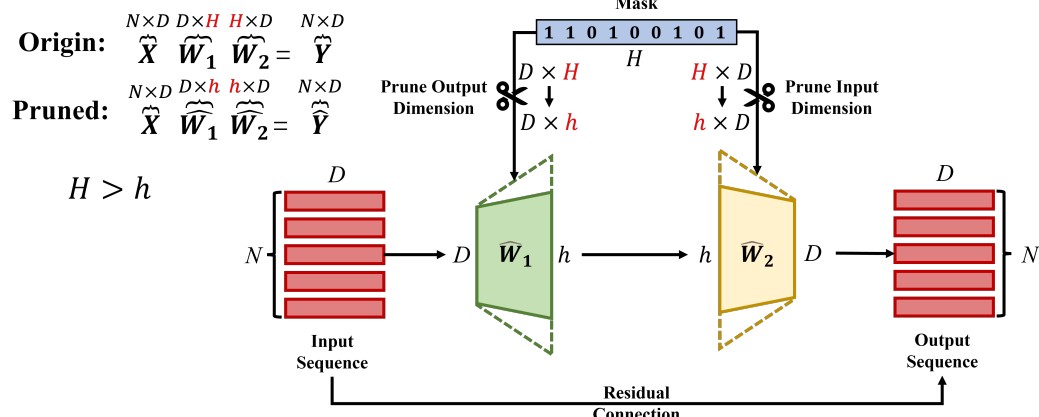

Figure A1: Output dimension is invariant for each block that might be used for residual connections, but instead prune the dimension of the intermediate hidden state.

## A.2 DETAILS ON HIDDEN STATES PRUNING FOR CHANNEL AND HEAD GRANULARITIES

We note that for pruning on the channel and head granularities, it must be guaranteed that the final output dimension for each block (*e.g.*, multi-head attention, MLP) should remain, so as to facilitate the residue connections (*e.g.*, additions) across blocks. We thus follow Ma et al. (2023); An et al. (2023) to prune the dimensions of the hidden states, while keeping the final output channels unchanged, ensuring that they can be added to the input through the residual connections. A conceptual figure illustrating this procedure is shown in Fig. A1.

## A.3 A REINFORCEMENT LEARNING PERSPECTIVE

Our formulation can also be interpreted from the dense-reward model-free reinforcement learning perspective. Particularly, the heavy LLM can be viewed as the agnostic and fixed environment.

In terms of the Markov Decision Process (MDP) (**action** $a$, **states** $s$, **state transition probability** $p$, **reward** $r$, **discount factor** $\gamma$), the environment takes the **action** $a$ sampled from the current `Bernoulli` *policy* $\pi$ to insert the binary masks for pruning, produces the **states** $s$ as the masked/pruned network deterministically (*i.e.*, the **state transition probability** $p$ is constantly 1), and generate the stepwise dense **reward** $r$ as the performance (*e.g.*, the cross-entropy loss) of the pruned LLM. Since our problem exhibits dense rewards, therefore the **discount factor** $\gamma$ is 1.

As a result, our policy to take actions, *i.e.*, the `Bernoulli` distribution to sample the binary masks, can be learned efficiently exploiting the *policy gradient estimator* (similar to REINFORCE), without back-propagating through the agnostic and fixed environment of the heavy LLM.

## A.4 THEORETICAL ANALYSIS OF MOVING AVERAGE BASELINE FOR POLICY GRADIENT

We give the theoretical analysis on the variance reduction technique by considering a general-purpose technique for reducing the variance of Monte Carlo method with the general problem $\mathbb{E}_{p(\mathbf{x};\theta)}[f(\mathbf{x})]$. We take a strategy that replacing the function $f(\mathbf{x})$ in the expectation by a substitute function $\tilde{f}(\mathbf{x})$ whose expectation $\mathbb{E}_{p(\mathbf{x};\theta)}[\tilde{f}(\mathbf{x})]$ is the same, but whose variance is lower. Given a function $h(\mathbf{x})$ with a known expectation $\mathbb{E}_{p(\mathbf{x};\theta)}[h(\mathbf{x})]$, we can easily construct such a substitute function along with the corresponding estimator as follows:

$$\tilde{f}(\mathbf{x}) = f(\mathbf{x}) - \beta(h(\mathbf{x}) - \mathbb{E}_{p(\mathbf{x};\theta)}[h(\mathbf{x})]), \ \bar{\eta}_N = \frac{1}{N}\sum_{n=1}^{N}\tilde{f}(\hat{\mathbf{x}}^n) = \bar{f} - \beta(\bar{h} - \mathbb{E}_{p(\mathbf{x};\theta)}[h(\mathbf{x})]).$$
(A3)

where $\hat{x}^n \sim p(\mathbf{x};\theta)$ and $\bar{f}$ and $\bar{h}$ are the sample averages. $\beta$ is a control coefficient and $h(\mathbf{x})$ is considered as control variate. We can show that if the variance of $h(\mathbf{x})$ is finite, the unbiasedness the estimator Eq. A3 is maintained, *e.g.*,

$$\mathbb{E}_{p(\mathbf{x};\theta)}[(\mathbf{x};\beta)] = \mathbb{E}[\bar{f} - \beta(\bar{h} - \mathbb{E}_{h(\mathbf{x})})] = \mathbb{E}[\bar{f}] = \mathbb{E}_{p(\mathbf{x};\theta)}[f(\mathbf{x})]. \quad (A4)$$

For the variance of the estimator (for N = 1), we have

$$\mathbb{V}[\tilde{f}] = \mathbb{V}[f(\mathbf{x}) - \beta(h(\mathbf{x}) - \mathbb{E}_{p(\mathbf{x};\theta)}[h(\mathbf{x})])] = \mathbb{V}[f] - 2\beta Cov[f,h] + \beta^2\mathbb{V}[h]. \quad (A5)$$

By minimizing Eq. A5 we can find that the optimal value of the coefficient is

$$\beta^* = \frac{\text{Cov}[f,h]}{\mathbb{V}[h]} = \sqrt{\frac{\mathbb{V}[f]}{\mathbb{V}[h]}}\text{Corr}(f,h), \quad (A6)$$

where we expressed the optimal coefficient in terms of the variance of $f$ and $h$, as well as in terms of the correlation coefficient $\text{Corr}(f,h)$. The effectiveness of the control variate can be characterized by the ratio of the variance of the control variate estimator to that of the original estimator: we can say our efforts have been effective if the ratio is substantially less than 1. Using the optimal control coefficient in Eq. A6, the potential variance reduction is

$$\frac{\mathbb{V}[\tilde{f}(\mathbf{x})]}{\mathbb{V}[f(\mathbf{x})]} = \frac{\mathbb{V}[f(\mathbf{x} - \beta(h(\mathbf{x}) - \mathbb{E}_{p(\mathbf{x};\theta)}[h(\mathbf{x})]))]}{\mathbb{V}[f(\mathbf{x})]} = 1 - \text{Corr}(f(\mathbf{x}), h(\mathbf{x}))^2. \quad (A7)$$

Therefore, as long as $f(\mathbf{x})$ and $h(\mathbf{x})$ are not uncorrelated, we can always obtain a reduction in variance using control variables. In practice, the optimal $\beta^*$ will not be known and so we will usually need to estimate it empirically.

In our problem formulation of structured pruning for LLMs, $\mathbb{E}_{p(\mathbf{m}|\mathbf{s})}\mathcal{L}(\mathcal{D};\mathbf{w}\odot\mathbf{m})\nabla_{\mathbf{s}}\log(p(\mathbf{m}|\mathbf{s}))$, is a score-function gradient estimator [1], in which $p(\mathbf{m}|\mathbf{s})$ is the `Bernoulli` distribution of each module of LLMs with $\mathbf{s}$ corresponds the $\theta$, $\mathbf{m}$ corresponds the $\theta$ and $\mathcal{L}(\mathcal{D};\mathbf{w}\odot\mathbf{m})\nabla_{\mathbf{s}}\log(p(\mathbf{m}|\mathbf{s}))$ corresponds $f(\mathbf{x})$ in the preliminary. To reduce the variance of a score-function gradient estimator, one simple and general way is to use the score function itself as a control variate, that is $h(\mathbf{m}) = \delta\nabla_{\theta}\log p(\mathbf{m}|\mathbf{s})$ and $\delta$ is an independent estimation of $\mathcal{L}(\mathcal{D};\mathbf{w}\odot\mathbf{m})$, since its expectation under the measure is zero, as

$$\mathbb{E}_{p(\mathbf{m}|\mathbf{s})}[\delta\nabla_{\mathbf{s}}\log p(\mathbf{m}|\mathbf{s})] = \delta\int p(\mathbf{m}|\mathbf{s})\frac{\nabla_{\mathbf{s}}p(\mathbf{m}|\mathbf{s})}{p(\mathbf{m}|\mathbf{s})}d\mathbf{m} = \delta\nabla_{\mathbf{s}}\int p(\mathbf{m}|\mathbf{s})d\mathbf{m} = \delta\nabla_{\mathbf{s}}1 = \mathbf{0}. \quad (A8)$$

Therefore, the estimator in Eq. A3 format is:

$$\bar{\eta}_N = \frac{1}{N}\sum_{n=1}^{N}(\mathcal{L}(\mathcal{D};\mathbf{w}\odot\mathbf{m}^{(n)}) - \beta\delta)\nabla_{\mathbf{s}}\log(p(\mathbf{m}^{(n)}|\mathbf{s})); \ \mathbf{m}^{(n)} \sim p(\mathbf{m}|\mathbf{s}), \quad (A9)$$

where $\mathbf{m}^{(n)}$ is the sampled mask of modules. In reinforcement learning, the term $\beta\delta$ is called a baseline Williams (1992) and has historically been estimated with a running average of the cost. Note that $\delta$ needs to be estimated, we choose moving average baseline in our method, which is a commonly used baseline in policy gradient estimation Zhao et al. (2011); Sehnke et al. (2010).

## A.5 STATISTICS OF THE TRAINING TIME & MEMORY, AND THE INFERENCE LATENCY

Our training times for channel and head pruning on LLaMA-2-7B and LLaMA-2-13B are 1.76 and 2.72 hours, respectively. Although our method is slower than metric-based methods such as Wanda-sp An et al. (2023), the trade-off is justified by the substantial performance enhancements delivered by our optimization-based approach.

The GPU memory requirements for channel and head pruning on LLaMA-2-7B and LLaMA-2-13B for our methods, as well as the representative metric-based method, *e.g.*, Wanda-sp, are illustrated in Table A1. We do not compare it to LLM-Pruner and SliceGPT because 1) the LLM-Pruner requires much more memory for back-propagation (therefore the authors also used the CPU memory), 2) the original implementation of SliceGPT also used both CPU and GPU memory for computations. Table A1 shows that our method exhibits a similar GPU memory requirement to the efficient Wanda-sp, as we only need the forward pass of the LLM. The slight additional memory required by our method comes from the need to store the `Bernoulli` parameters $\mathbf{s}$ and the sampled masks $\mathbf{m}$.

Table A1: Memory requirements (GB) for channel and head pruning on LLaMA-2-7B/13B.

| Method | 7B | | 13B | |
|---|---|---|---|---|
| | Min | Max | Min | Max |
| Wanda-sp | 17.5 | 20.3 | 29.5 | 36.9 |
| Ours | 18.2 | 19.5 | 34.1 | 35.8 |

We note that for the same pruning rate (*i.e.*, similar remaining #Params), the inference latencies of pruned models from different structural pruning methods are expected to be comparable, as the inference latency is mainly affected by the #Params given the same architecture. We validate this in Table A2. Table A2 demonstrates that, given the same pruning rates, our pruned model has very much close #Params, memory, and inference latencies to that pruned by LLM-Pruner, while our perplexity significantly outperformed all the counterparts. We note that under the same pruning rates, SliceGPT often possesses different (higher) #Params, memory, and inference latencies than our method and LLM-Pruner, potentially because SliceGPT alters the network structure during the pruning.

Table A2: #Params, memory requirements, latency and perplexity on WikiText2 dataset of LLaMA-2-7B. The experiments are conducted on NVIDIA A100 40G, with 2048 sequence length and 4 batch size for sufficient GPU utilization.

| Method | PruneRate | #Params (B) | Memory (MiB) | Latency (s) | Perplexity |
|---|---|---|---|---|---|
| LLM-Pruner | | 4.837 | **9290.54** | 53.53 | 27.13 |
| SliceGPT | 30% | 5.293 | 10181.81 | 50.24 | 22.29 |
| Ours | | **4.796** | 9338.24 | **46.94** | **12.68** |
| LLM-Pruner | | 4.197 | **8069.55** | **36.75** | 53.21 |
| SliceGPT | 40% | 4.501 | 8826.01 | 46.84 | 39.21 |
| Ours | | **4.149** | 8096.25 | 42.85 | **15.95** |
| LLM-Pruner | | 3.539 | **6815.05** | **31.49** | 171.57 |
| SliceGPT | 50% | 3.730 | 7274.01 | 41.73 | 65.92 |
| Ours | | **3.500** | 6880.92 | 34.62 | **27.63** |

## A.6 PERFORMANCE AFTER PRUNING AND (THEN) FINETUNING

We note that after pruning, it becomes affordable to finetune a smaller pruned model. Therefore, following the idea from Ma et al. (2023), we finetune the post-pruning model *w.r.t.* the perplexity with LoRA Hu et al. (2022). Specifically, We utilize 4k samples from the Alpaca Taori et al. (2023) dataset, which has a sequence length of 1024. For all weight fine-tuning experiments, we use *lora_r* = 16, *lora_alpha* = 10, and use default values for all other hyperparameters in the HuggingFace PEFT package Mangrulkar et al. (2022).

The cross-dataset performance on WikiText of the post-pruning fine-tuned model for LLaMA-2-7B and LLaMA-3-8B is illustrated in Tables A3 and A4, which demonstrate that our method achieves consistently superior performance before and after fine-tuning. Compared with the pre-finetuned model, the performance of most post-finetuned models shows significant improvements, and our models remain the best for most cases after funetuning, which validates our potential for narrowing the performance gap after pruning and for being applicable in practical use.

Table A3: Perplexity (PPL) and Accuracies (%) of LLaMA-2-7B for 5 zero-shot tasks with pruning rates from 30% to 50% after weight fine-tuning on Alapca dataset.

| Method | PruneRate | PPL ↓ | PIQA | HellaSwag | WinoGrande | ARC-e | ARC-c | Average |
|---|---|---|---|---|---|---|---|---|
| Dense | 0% | 12.19 | 78.02 | 57.17 | 68.43 | 76.30 | 43.51 | 64.69 |
| LLM-Pruner | | 33.45 | 74.10 | 46.61 | 58.17 | 64.31 | 33.62 | 55.36 |
| SliceGPT | | 78.59 | 74.70 | **64.29** | 61.96 | 57.49 | 36.69 | 59.03 |
| Bonsai | 30% | 33.23 | 75.03 | 49.69 | 62.19 | 67.34 | 32.25 | 57.30 |
| Wanda-sp | | 32.01 | 73.88 | 50.08 | 62.19 | 67.09 | 34.47 | 57.54 |
| Ours | | **25.34** | **76.01** | 51.80 | **64.33** | 67.93 | 36.86 | **59.39** |
| LLM-Pruner | | 40.21 | 70.29 | 40.45 | 53.04 | 53.03 | 27.30 | 48.82 |
| SliceGPT | | 175.67 | 65.29 | **56.77** | **60.06** | 42.68 | **31.74** | 51.31 |
| Bonsai | 40% | 44.71 | 72.36 | 45.10 | 58.80 | 59.64 | 30.03 | 53.19 |
| Wanda-sp | | 43.71 | 70.40 | 42.73 | 52.72 | 57.24 | 29.95 | 50.61 |
| Ours | | **29.43** | **72.74** | 45.75 | 55.72 | **61.36** | 31.06 | **53.33** |
| LLM-Pruner | | 44.83 | **67.30** | 35.47 | 51.93 | 48.23 | 21.84 | 44.95 |
| SliceGPT | | 296.97 | 58.65 | **46.83** | **55.09** | 36.99 | **28.33** | 45.18 |
| Bonsai | 50% | 62.95 | 66.70 | 40.16 | 54.30 | 49.83 | 26.53 | **47.50** |
| Wanda-sp | | 110.12 | 63.27 | 32.71 | 52.72 | 43.48 | 20.73 | 42.58 |
| Ours | | **39.46** | 67.03 | 36.42 | 52.41 | **50.17** | 24.15 | 46.04 |

Table A4: Perplexity (PPL) and Accuracies (%) of LLaMA-3-8B for 5 zero-shot tasks with pruning rates from 30% to 50% after weight fine-tuning on Alapca dataset.

| Method | PruneRate | PPL ↓ | PIQA | HellaSwag | WinoGrande | ARC-e | ARC-c | Average |
|---|---|---|---|---|---|---|---|---|
| Dense | 0% | 14.13 | 79.71 | 60.19 | 72.61 | 80.09 | 50.34 | 68.59 |
| LLM-Pruner | | 35.11 | **74.64** | 46.93 | 60.22 | **66.16** | 34.13 | 56.42 |
| SliceGPT | | 226.39 | 70.29 | **56.47** | 60.06 | 53.20 | 34.81 | 54.97 |
| Bonsai | 30% | 42.59 | 71.87 | 45.17 | 59.51 | 66.50 | **36.52** | 55.91 |
| Wanda-sp | | 38.04 | 70.84 | 44.11 | 59.43 | 62.96 | 34.04 | 54.28 |
| Ours | | **33.91** | 74.48 | 46.62 | **63.69** | 65.70 | 34.30 | **56.96** |
| LLM-Pruner | | 47.83 | **71.54** | 40.71 | 55.40 | **62.16** | 28.92 | 51.75 |
| SliceGPT | | 523.05 | 63.66 | **42.75** | 53.12 | 41.88 | 27.65 | 45.81 |
| Bonsai | 40% | 57.31 | 69.58 | 39.47 | 53.98 | 57.24 | 28.67 | 49.79 |
| Wanda-sp | | 56.32 | 65.18 | 36.33 | 54.77 | 51.56 | 24.32 | 46.43 |
| Ours | | **47.28** | 70.56 | 41.09 | **59.98** | 59.97 | **29.01** | **52.12** |
| LLM-Pruner | | 68.14 | **67.95** | 35.81 | 53.12 | **53.91** | 26.36 | 47.43 |
| SliceGPT | | 963.42 | 60.83 | **37.04** | 52.25 | 37.21 | 25.26 | 42.52 |
| Bonsai | 50% | 88.72 | 62.89 | 34.84 | 52.80 | 47.73 | 24.15 | 44.48 |
| Wanda-sp | | 84.53 | 61.42 | 32.12 | 52.72 | 41.83 | 21.07 | 41.83 |
| Ours | | **67.48** | 67.08 | 35.84 | **54.38** | 53.54 | **26.45** | **47.46** |

## A.7 ZERO-SHOT PERFORMANCE ON LLAMA-2-7B

We validate the zero-shot performance of LLaMA-2-7B with prune rates from 30% to 50%, shown in Table A5. We note that the overall performance is in general superior to the baselines, though using only the C4 dataset for pruning might introduce a negative influence on some particular cross-dataset zero-shot tasks such as WinoGrande Sakaguchi et al. (2021) and Hellaswag Zellers et al. (2019). We discuss this in our limitations.

## A.8 PRUNING RESULTS ON MISTRAL-7B-INSTRUCT-V0.3

To further validate the performance of the proposed method on more LLMs, we additionally perform experiments on Mistral-7B-Instruct-v0.3 Jiang et al. (2023), which takes the C4 dataset as calibration and evaluates on the WikiText2 dataset (*e.g.*, cross-dataset setting, the same as those in our Fig. 3).

Table A5: Perplexity (PPL) and accuracies (%) of LLaMA-2-7B for 5 zero-shot tasks with pruning rates from 30% to 50%.

| Method | PruneRate | PPL ↓ | PIQA | HellaSwag | WinoGrande | ARC-e | ARC-c | Average |
|---|---|---|---|---|---|---|---|---|
| Dense | 0% | 12.19 | 78.02 | 57.17 | 68.43 | 76.30 | 43.51 | 64.69 |
| LLM-Pruner | | 38.94 | 71.81 | 43.64 | 54.06 | 63.42 | 30.30 | 52.64 |
| SliceGPT | | 40.40 | 72.31 | **60.11** | **63.22** | 53.10 | 32.00 | 56.15 |
| Bonsai | 30% | 39.01 | 73.94 | 47.05 | 60.93 | 59.93 | 30.37 | 54.44 |
| Wanda-sp | | 49.13 | 71.60 | 46.62 | 60.30 | 63.01 | 34.04 | 55.11 |
| Ours | | **28.18** | **75.41** | 50.34 | 61.60 | **66.03** | **35.58** | **57.79** |
| LLM-Pruner | | 68.48 | 67.52 | 35.76 | 51.70 | 48.31 | 24.65 | 45.59 |
| SliceGPT | | 73.76 | 65.40 | **48.91** | **60.38** | 42.13 | 26.88 | 48.74 |
| Bonsai | 40% | 69.18 | 68.44 | 40.63 | 55.41 | 48.11 | 26.19 | 47.75 |
| Wanda-sp | | 78.45 | 64.63 | 35.65 | 52.17 | 48.11 | 25.51 | 45.21 |
| Ours | | **39.81** | **71.11** | 42.44 | 55.72 | **56.94** | **28.50** | **50.94** |
| LLM-Pruner | | 190.56 | 59.52 | 29.74 | 50.11 | 36.48 | 21.84 | 39.54 |
| SliceGPT | | 136.33 | 59.47 | **37.96** | **56.27** | 33.63 | **22.78** | **42.02** |
| Bonsai | 50% | 216.85 | 59.52 | 32.63 | 53.12 | 33.54 | 22.61 | 40.28 |
| Wanda-sp | | 206.94 | 54.30 | 26.81 | 52.80 | 29.12 | 19.20 | 36.45 |
| Ours | | **65.21** | **61.80** | 30.94 | 52.64 | **40.11** | 20.47 | 41.19 |

We note that the original implementations of SliceGPT Ashkboos et al. (2024) and Bonsai Dery et al. (2024) were based on LLaMA-2, which do not trivially adapt to the Mistral model directly, therefore we exclude SliceGPT and Bonsai for comparison.

The results, including both perplexity and the zero-shot performance, on Mistral-7B-Instruct-v0.3 in Table A6 demonstrate the consistent superiority of our method across various LLMs.

Table A6: Perplexity (PPL) and accuracies (%) of Mistral-7B-Instruct-v0.3 for 5 zero-shot tasks with pruning rates from 30% to 50%.

| Method | PruneRate | PPL ↓ | PIQA | HellaSwag | WinoGrande | ARC-e | ARC-c | Average |
|---|---|---|---|---|---|---|---|---|
| Dense | 0% | 12.70 | 81.77 | 64.84 | 74.51 | 84.22 | 57.34 | 72.54 |
| LLM-Pruner | | **30.32** | 69.58 | 41.52 | 57.77 | 53.99 | 28.58 | 50.29 |
| Wanda-sp | 30% | 47.30 | 75.68 | 49.94 | 62.35 | 64.90 | 36.60 | 57.89 |
| Ours | | 31.87 | **76.49** | **52.69** | **64.48** | **67.76** | **36.77** | **59.64** |
| LLM-Pruner | | 49.30 | 65.18 | 34.79 | 52.80 | 46.42 | 23.89 | 44.62 |
| Wanda-sp | 40% | 76.45 | 68.01 | 38.75 | 52.64 | 52.36 | 26.28 | 47.61 |
| Ours | | **43.02** | **68.61** | **40.80** | **56.67** | **54.80** | **27.82** | **49.74** |
| LLM-Pruner | | 86.24 | 61.31 | 30.64 | 49.64 | 37.67 | 22.52 | 40.36 |
| Wanda-sp | 50% | 407.33 | 56.69 | 29.08 | 49.25 | 32.36 | 21.59 | 37.79 |
| Ours | | **74.25** | **65.18** | **35.02** | **51.06** | **48.15** | **22.61** | **44.40** |

## A.9 RANDOM ERROR-BAR STATISTIC

The standard deviation statistics of our method are shown in Table A7. Theoretically, the variance is induced by stochastic sampling from `Bernoulli` distribution in the policy gradient optimization if the initialization is fixed. Therefore, we fixed the initialization as Wanda-sp to calculate the standard deviation of the proposed method. Experiments of head and channel pruning, along with layer pruning, are executed using LLaMA-2-7B for 10 run trials. Table A7 shows that our method possesses a reasonable standard deviation.

Table A7: Mean and standard deviation of our method for LLaMA-2-7B.

| Granularity | PruneRate | | |
|---|---|---|---|
| | 30% | 40% | 50% |
| Head and Channel | 28.18 ± 1.83 | 39.81 ± 1.41 | 65.21± 2.52 |
| Layer | 23.20 ± 0.67 | 38.26 ± 2.68 | 104.37 ± 1.05 |

### A.10 ABLATIONS ON THE MOVING AVERAGE BASELINE FOR POLICY GRADIENT

We conduct experiments on pruning channels and heads of LLaMA-2-7B/13B with/without the *Moving Average Baseline* in policy gradient. Table A8 illustrates the effectiveness of the moving average baseline in the policy gradient estimator for our proposed pruning method.

Table A8: Ablations on the proposed Moving Average Baseline (MAB) in the policy gradient estimator for Channels and heads pruning on LLaMA-2-7B/13B.

| Method | PruneRate | LLaMA-2-7B | LLaMA-2-13B |
|---|---|---|---|
| w/o MAB | 30% | 32.53 | 24.73 |
| with MAB | | **28.18** | **21.99** |
| w/o MAB | 40% | 60.99 | 64.34 |
| with MAB | | **39.81** | **31.52** |
| w/o MAB | 50% | 69.47 | 185.87 |
| with MAB | | **65.21** | **52.23** |

Moreover, we also tested all the hyper-parameters, *e.g.*, the window size and the mask sampling times ($T$ and $N_s$ in Eq. (A9)). The results in Table A9 demonstrate that, being different from with vs. without moving average baseline, small $T$ and $N_s$ can already offer promising performance, further increasing them only produces marginal improvement. In other words, our method is robust to those hyper-parameter values. Considering the computational overhead, we choose small $T = 5$ and $N_s = 2$ throughout our entire experiments.

Table A9: Ablation on the hyperparameters of the moving average baseline, *i.e.*, different window sizes $T$ and mask sampling times $N_s$. Perplexity is tested on WikiText2 dataset of LLaMA-2-13B with 30% pruning rate. The hyper-parameter values used by main results are denoted with $\star$.

| Hyper-params | $T$ | | | $N_s$ | | |
|---|---|---|---|---|---|---|
| | 3 | 5$\star$ | 7 | 2$\star$ | 3 | 4 |
| Perplexity | 21.23 | 21.99 | **20.08** | 21.99 | 21.71 | **21.37** |

### A.11 ABLATIONS ON PROJECTION STRATEGY FOR INITIALIZATION: FROM METRIC TO PROBABILITY

As the initialization of our `Bernoulli` policy should be probabilistic values between 0 and 1, but the metrics calculated by the metric-based methods Sun et al. (2023); An et al. (2023); Ma et al. (2023) may not hold this range, we thus need to project those metric values to [0, 1] as our initialization. We introduce two projection strategies from metric values $\mathbf{m}$ to probabilities $\mathbf{s}$. The first is called *Sigmoid-Norm* strategy, which is applied in our main experiments:

$$\mathbf{s} = \text{sigmoid}(\text{Norm}(\mathbf{x})) \qquad (A10)$$

where $\text{Norm}(\cdot)$ is used to linearly normalize the input to a Gaussian distribution with 0 mean and unit variance, then $sigmoid(\cdot)$ is used to transform the input to [0, 1].

An alternative second strategy is named *Score-Const*. It straightforwardly sets the mask 1 from metric-based methods as a constant $c$, and mask 0 as $1 - c$:

$$s_i = \begin{cases} c, & \text{if } m_i = 1, \\ 1 - c, & \text{if } m_i = 0, \end{cases} \qquad (A11)$$

The constant $c$ is set to 0.8 in the following experiments, indicating that the initialized `Bernoulli` probability of the remaining modules is 0.8 and those to be pruned is 0.2.

The results of different projection strategies on LLaMA-2-7B/13B are detailed in Table A10, which shows that the *Sigmoid-Norm* projection outperforms its *Score-Const* counterpart for most cases. It may be because the order-preserving projection strategy of *Sigmoid-Norm* preserves more information about relative importance among modules, and therefore benefits the optimization.

### A.12 MORE ABLATIONS WITH DIFFERENT INITIALIZATIONS

**Progressive Pruning with Random (Random-Progressive) Initialization.** Our progressive pruning with random initialization is inspired by the facts that 1) the *continous* `Bernoulli` probability

Table A10: Results with *different projection strategies* for pruning heads, channels, and layers on LLaMA-2-7B/13B. Initialization metrics are from Wanda-sp for heads/channels and Layerwise-PPL for layers.



(a) Channels and Heads Pruning.

| Method | Sparsity | 7B | 13B |
|--------|----------|-----|-----|
| Sigmoid-Norm | 30% | **28.18** | **21.99** |
| Score-Const | | 32.25 | 25.38 |
| Sigmoid-Norm | 35% | **32.52** | **26.27** |
| Score-Const | | 40.61 | 40.51 |
| Sigmoid-Norm | 40% | **39.81** | **31.52** |
| Score-Const | | 44.46 | 52.10 |
| Sigmoid-Norm | 45% | **52.07** | **40.99** |
| Score-Const | | 65.31 | 61.04 |
| Sigmoid-Norm | 50% | **65.21** | **52.23** |
| Score-Const | | 77.07 | 88.72 |

(b) Layer Pruning.

| Method | Sparsity | 7B | 13B |
|--------|----------|-----|-----|
| Sigmoid-Norm | 30% | **23.20** | 21.93 |
| Score-Const | | 25.32 | **19.31** |
| Sigmoid-Norm | 35% | 33.27 | 26.46 |
| Score-Const | | **31.37** | **23.40** |
| Sigmoid-Norm | 40% | **38.26** | 30.99 |
| Score-Const | | 42.30 | **29.25** |
| Sigmoid-Norm | 45% | 69.23 | **39.26** |
| Score-Const | | **63.91** | 39.50 |
| Sigmoid-Norm | 50% | **104.37** | 69.92 |
| Score-Const | | 135.51 | **54.37** |



learned by our method indicates the importance of the corresponding module, therefore the *continous* probability scores from a low pruning rate (*e.g.*, 10%) encodes fatal information and can be naturally used as the initialization for a higher pruning rate (*e.g.*, 15%); and 2) the LLMs is likely to exhibit large redundancy when the pruning rate is extremely low (*e.g.*, 5%), thus random initialization will not significantly degrade the pruning performance (compared to a carefully chosen metric-based pruning initialization) given an extremely low pruning rate such as 5%. Therefore, to validate our method without a prior metric-based initialization, we propose a progressive pruning strategy, by starting from 5% pruning rate with random initialization and progressively pruning rate to 50% by a step size of 5%. We train this strategy with each pruning rate for 1/3 epoch to maintain the efficiency.

In addition to Table 2 of Sect. 5.1, Table A11 shows the *layer* pruning results with different initializations on LLaMA-2-7B.

Table A11: Layer pruning results with *different initializations* using LLaMA-2-7B. **Bold** and Underscored denote the first and second best results, respectively.

| Method | PruneRate | Perplexity | PruneRate | Perplexity | PruneRate | Perplexity |
|--------|-----------|-----------|-----------|-----------|-----------|-----------|
| Layerwise-PPL | 30% | 24.83 | 40% | 41.45 | 50% | 126.08 |
| Ours (Random Init) | 30% | 26.65 | 40% | 42.76 | 50% | 125.20 |
| Ours (Random-Prog. Init) | | 30.05 | | 38.28 | | 111.87 |
| Ours (Layerwise-PPL Init) | 30% | **23.20** | 40% | **38.26** | 50% | **104.37** |

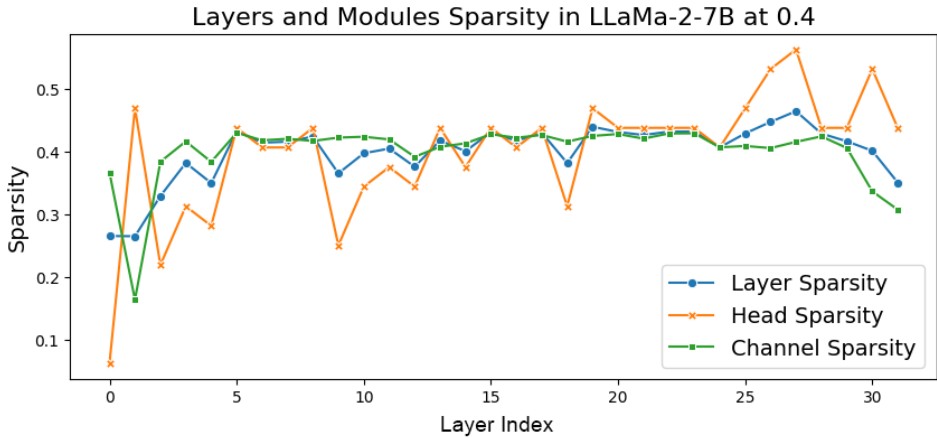

Figure A2: Channels, heads, and layers sparsities on *LLaMA-2-7B*.

## A.13 MORE ABLATIONS OF THE POST-PRUNING MODULES ON LLaMA-2-7B

Besides the LLaMA-2-13B results in Fig. 6 of Sect. 5.3, the channels, heads, and layers sparsities on LLaMA-2-7B with channels and heads pruning are shown in Fig. A2, which illustrates a similar observation as Fig. 6.

## A.14 HAMMING DISTANCE ON THE MASKS GENERATED FROM DIFFERENT METHODS

The normalized Hamming distances of pruning masks between our method and LLM-Pruner, Bonsai, Wanda-sp on LLaMA-2-7B with 30% pruning rate are shown in Fig. A3, where we calculated the normalized Hamming distances according to each layer. Note that SliceGPT altered the architecture of the LLM to prune, making the hamming distance analysis less reasonable. Figure A3 demonstrates that the learned masks are different from our methods and our metric-based counterparts.

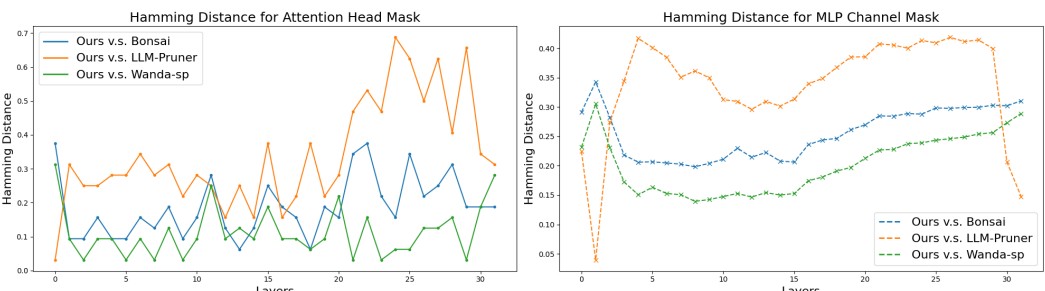

Figure A3: Normalized Hamming distance of the mask generated from different methods on *LLaMA-2-7B* with 30% pruning rate.

## A.15 PERFORMANCE USING THE SAME AMOUNT OF CALIBRATION DATA

In our main text, we used larger calibration data than our metric-based counterpart methods because: 1) **Data Availability**: Thanks to the self-supervised learning paradigm in LLM training, a large amount of unlabeled text data is easy to acquire nowadays; and 2) **Cross-dataset Experiments**: The cross-dataset performance of our method was extensively validated throughout all of our experiments. This demonstrates an alleviated requirement of the calibration data for pruning, i.e., the calibration data and the evaluation data do not need to originate from the same dataset (e.g., C4 vs. WikiText).

It is interesting to see how the pruning methods perform using the same amount of calibration data. Specifically, we use the calibration data of **128 samples with 2048 sequence length** from the C4 dataset for all the methods, and perform the **LLaMA-2-7B** experiment with **20% and 30%** sparsity.

The results are shown in Table A12, which demonstrates that our method consistently outperforms counterpart methods for the majority of cases when using the same amount of calibration data.

Table A12: Perplexity (PPL) and accuracies (%) of LLaMA-2-7B for 5 zero-shot tasks of 20% to 30% pruning rates, with **128 samples with 2048 sequence length** from the C4 dataset for calibration

| Method | PruneRate | PPL ↓ | PIQA | HellaSwag | WinoGrande | ARC-e | ARC-c | Average |
|--------|-----------|-------|------|-----------|------------|-------|-------|---------|
| Dense | 0% | 12.19 | 78.02 | 57.17 | 68.43 | 76.30 | 43.51 | 64.69 |
| SliceGPT | | 24.87 | 74.92 | 49.91 | 66.22 | 69.11 | 35.32 | 59.10 |
| Wanda-sp | 20% | 23.08 | 77.09 | **54.34** | 65.9 | 71.21 | **40.27** | 61.76 |
| Bonsai | | 23.03 | 76.82 | 53.1 | 64.25 | 71.17 | 39.85 | 61.04 |
| Ours | | **19.61** | **77.09** | 53.45 | **66.38** | **72.39** | 40.02 | **61.87** |
| SliceGPT | | 40.96 | 71.71 | 44.58 | **64.80** | 60.73 | 30.20 | 54.40 |
| Wanda-sp | 30% | 42.96 | 74.59 | 48.43 | 59.12 | 63.47 | 34.30 | 55.98 |
| Bonsai | | 48.30 | 72.85 | 48.25 | 57.77 | 63.80 | 33.87 | 55.31 |
| Ours | | **27.13** | **75.79** | **49.00** | 62.27 | **65.36** | **34.56** | **57.40** |

