# OpenReview forum: "Bypass Back-propagation: Optimization-based Structural Pruning for Large Language Models via Policy Gradient"
_ICLR.cc/2025/Conference — Submitted to ICLR 2025_

### Official Review · Reviewer_pAjX · 2024-10-29

**Soundness:** 2
**Presentation:** 2
**Contribution:** 2
**Rating:** 3
**Confidence:** 4

**Summary:**

This paper proposes a novel optimization-based structural pruning method for Large Language Models (LLMs) that avoids expensive back-propagation through the LLM. The key idea is to formulate pruning as learning binary masks sampled from underlying Bernoulli distributions. By decoupling the Bernoulli parameters from the LLM loss, the method enables efficient optimization using policy gradient estimation that only requires forward passes. The approach supports different structural granularities (channels, heads, layers) and enables global/heterogeneous pruning across the model. Extensive experiments on various LLMs demonstrate strong performance while maintaining efficiency.

**Strengths:**

- Clear writing and good organization of the paper
- The paper presents a novel methods for structural pruning that leverages policy gradient estimation, bypassing the need for back-propagation through large models.
- The method supports multiple pruning granularities (channels, heads, layers) and can be initialized using metric-based methods, showcasing its adaptability and potential for widespread application.

**Weaknesses:**

1. **Lack of Comparison with Recent Baselines**:
   The paper does not compare the proposed method with recent pruning techniques such as Shortened LLaMA ([arXiv:2402.02834](https://arxiv.org/abs/2402.02834)) and SLEB ([arXiv:2402.09025](https://arxiv.org/abs/2402.09025)), which were published prior to ShortGPT ([arXiv:2403.03853](https://arxiv.org/abs/2403.03853)) and Gromov et al.'s work ([arXiv:2403.17887](https://arxiv.org/abs/2403.17887)). Additionally, MKA ([arXiv:2406.16330](https://arxiv.org/abs/2406.16330)) is also not included. Including comparisons with these methods on benchmarks like MMLU and GSM8K, using metrics like accuracy and perplexity, would provide a more comprehensive evaluation of the method's performance relative to the latest advancements in the field.

**Questions:**

1. Does the paper include accuracy (ACC) results on more advanced benchmarks like MMLU or GSM8K? If not, could the authors provide such evaluations to demonstrate the method's effectiveness on tasks that require higher reasoning capabilities?

2. The proposed method exhibits longer training times compared to metric-based pruning methods. Are there potential optimizations or strategies that the authors are considering to reduce the training duration without compromising performance?

---

> ### Author Response · Authors · 2024-11-26
> **[Second Round Response] Part 1**
>
> The authors respectfully disagree with the comments from Reviewer pAjX. \
> Reviewer pAjX gave an extremely low score of 3 justified by only one Weakness: "*Lack of Comparison with Recent Baselines*", which **violates the ICLR 2025 instruction** and **contradicts both comments from the same reviewer and strengths from Reviewers 1Xw7 and iXeF**. \
> **The second round review from Reviewer pAjX is erroneous**.
>
> ***
>
> [**Violation of ICLR 2025 Review Instruction**]
>
> 1. Per the latest instructions from PCs: "*Reviewers are instructed to not ask for significant experiments, and area chairs are instructed to discard significant requests from reviewers.*" \
> **Performing novel experiments on MMLU and GSM8K qualifies as *significant***, because:
>
> 1a. **Nether** MMLU **nor** GSM8K remains a standard benchmark for LLM pruning evaluation. GSM8K is absent from most state-of-the-art pruning methods [1-22] (including the five papers suggested by Reviewer pAjX [18-22]), and MMLU is only evaluated in ShortGPT [20], Gromov et al. [21], and MKA [22], however, none of [20-22] have released their code yet.
>
> 1b. As a consequence of 1a, it is difficult to find an LLM pruning method releasing the evaluation code on MMLU or GSM8K, nor were the authors specifically instructed by Reviewer pAjX about which setting is desirable.
>
> 1c. Despite 1a and 1b well justified that novel experiments on MMLU and GSM8K are **significant**, the author still made every effort to perform this for a rebuttal with the best quality. \
> As a consequence of 1b, the authors implemented the evaluation code for MMLU and GSM8K themselves. During the intense rebuttal period, the authors adhered to the same challenging experimental settings as those in their main experiments **for all the methods**, including:
> - Severe pruning rates of 30% to 50%,
> - Cross-dataset pruning + evaluation with calibration using an independent dataset (C4),
> - Keeping model weights unchanged/un-finetuned.
> These challenging settings result in less satisfactory performance on MMLU and GSM8K **for all the methods**.
>
> ***
>
> [**Contradictions/Errors in (Original, Round 1) Reviews**]
>
> 2. [**Contradictions**] The only weakness indicated by Reviewer pAjX is "*Lack of Comparison with Recent Baselines*". Despite that the proposed method achieves the best results for the majority cases over various SOTAs on both Perplexity and the most widely used zero-shot tasks of PIQA, HellaSwag, WinoGrande, ARC-e, ARC-c, this review contradicts with:
>
> 2a. The comments in the Summary Section from the same reviewer (Reviewer pAjX): "**Extensive experiments** on **various LLMs** demonstrate strong performance while maintaining efficiency".
>
> 2b. Strength from Reviewer 1Xw7: "The authors have conducted **extensive experimental evaluation** and **compare with many existing baseline methods** for structural pruning.
>
> 2c. Strength from Reviewer iXeF: "**Extensive experimental validation** across **multiple LLM architectures** and **datasets**".
>
> ***
>
> 3. [**Errors**] Five methods [18-22] were suggested for comparison, however,
>
> 3a. **1 out of 5** (i.e., [18], LayerwisePPL) has already been extensively compared in the original manuscript in Table 2 (now Figure 4 in the updated manuscript) and Table A11.
>
> 3b. **3 out of 5** (i.e., [20-22]) have not yet released their codes, making meaningful comparisons infeasible.
>
> 3c. Only [19] is applicable for comparison. The authors conducted this comparison during the rebuttal phase, which demonstrated that **the proposed method performs comparably to [19] while offering more flexible module pruning capabilities** (e.g., prune heads of multi-head attention, and prune channels of MLP), which [19] does not support.
>
> ***
>
> [**Errors in (Round 2) Reviews**]
>
> 4. ["*Performance is one-sided*"] Despite the challenges in 1c, the proposed method achieves the best results (among various SOTAs) in most cases on MMLU and GSM8K for head, channel, and layer pruning. If Reviewer pAjX finds this "*one-sided*", it would be necessary to consult ACs and other reviewers if the importance of LLM pruning as a research area should be reassessed.
>
> 5. ["*Performance is lower than dense model*"] Professional researchers in the network pruning field understand that it is inappropriate to directly compare pruned performance with a 100% dense model (upper bound), particularly with severe pruning rates of 30% to 50% without weights finetuning.
>
> 6. ["*Performance is lower than dense model*"] If insisting on comparing with the dense model, the vanilla LLaMA-3-8B achieves 79.6 on GSM8K (8-shot) is **NOT TRUE**. 79.6 is from [LLaMA-3-8B-Instruction-Tuned](https://github.com/EleutherAI/lm-evaluation-harness/issues/1896).
>
>
> ***
>
> The authors reply here solely for 100% transparency, the authors would like to draw the attention of ACs and other reviewers for this reply. The authors will not engage in discussion here further if consistently unprofessional reviews are received from Reviewer pAjX.
>
> Authors of Submission 745

---

> ### Author Response · Authors · 2024-11-26
> **[Second Round Response] Part 2**
>
> **References:**
>
> 1. Ashkboos et al. SliceGPT: Compress Large Language Models by Deleting Rows and Columns. ICLR, 2024
> 2. Zhang et al. Plug-and-Play: An Efficient Post-training Pruning Method for Large Language Models. ICLR, 2024
> 3. Zhang et al. Dynamic Sparse No Training: Training-Free Fine-Tuning for Sparse LLMs. ICLR, 2024
> 4. Yin et al. Outlier Weighed Layerwise Sparsity (OWL): A Missing Secret Sauce for Pruning LLMs to High Sparsity. ICML, 2024
> 5. Ma et al. LLM-Pruner: On the Structural Pruning of Large Language Models. NeurIPS, 2023
> 6. Frantar et al. SparseGPT: Massive Language Models Can Be Accurately Pruned in One-Shot. ICML, 2023
> 7. Wei et al. Assessing the Brittleness of Safety Alignment via Pruning and Low-Rank Modifications. ICML, 2024
> 8. Ko et al. NASH: A Simple Unified Framework of Structured Pruning for Accelerating Encoder-Decoder Language Models. Findings of EMNLP, 2023
> 9. An et al. Fluctuation-based Adaptive Structured Pruning for Large Language Models. AAAI, 2024
> 10. Zeng et al. Multilingual Brain Surgeon: Large Language Models Can Be Compressed Leaving No Language Behind. COLING, 2024
> 11. Shao et al. One-Shot Sensitivity-Aware Mixed Sparsity Pruning for Large Language Models. ICASSP, 2024
> 12. van der Ouderaa et al. The LLM Surgeon. arXiv:2312.17244, 2023
> 13. Boža. Fast and Effective Weight Update for Pruned Large Language Models. arXiv:2401.02938, 2024
> 14. Chen et al. LoRAShear: Efficient Large Language Model Structured Pruning and Knowledge Recovery. arXiv:2310.18356, 2023
> 15. Li et al. E-Sparse: Boosting the Large Language Model Inference through Entropy-based N:M Sparsity. arXiv:2310.15929, 2023
> 16. Xu et al. BESA: Pruning Large Language Models with Blockwise Parameter-Efficient Sparsity Allocation. arXiv:2402.16880, 2024
> 17. Das et al. Beyond Size: How Gradients Shape Pruning Decisions in Large Language Models. arXiv:2311.04902, 2023
> 18. Kim et al. Shortened LLaMA: A Simple Depth Pruning for Large Language Models. ICLR Workshop, 2024
> 19. Song et al. SLEB: Streamlining LLMs through Redundancy Verification and Elimination of Transformer Blocks. ICML 2024
> 20. Men et al. ShortGPT: Layers in Large Language Models are More Redundant Than You Expect. arXiv:2403.03853, 2024
> 21. Gromov et al. The Unreasonable Ineffectiveness of the Deeper Layers. arXiv:2403.17887, 2024
> 22. Liu et al. Pruning via Merging: Compressing LLMs via Manifold Alignment Based Layer Merging. arXiv:2406.16330, 2024

---

### Official Review · Reviewer_iXeF · 2024-10-30

**Soundness:** 3
**Presentation:** 2
**Contribution:** 3
**Rating:** 6
**Confidence:** 3

**Summary:**

This paper presents a novel approach to structural pruning of LLMs that bridges the gap between optimization-based and metric-based pruning methods. The key innovation lies in the use of Bernoulli distributions to sample binary pruning masks, coupled with a policy gradient estimator that eliminates the need for back-propagation through the LLM. This is a clever solution that maintains the benefits of optimization-based approaches while achieving the efficiency typically associated with metric-based methods.

**Strengths:**

- The proposed method is both theoretically sound and practically implementable, requiring only forward passes through the LLM during optimization.
(1) The efficiency claims are impressive (2.7 hours, 35GB memory for 13B models on a single A100)
(2) The approach is flexible, supporting multiple structural granularities (channels, heads, and layers)
(3) The method automatically handles heterogeneous pruning across layers, addressing a key limitation of existing approaches
(4) Extensive experimental validation across multiple LLM architectures and datasets

**Weaknesses:**

The paper could benefit from more detailed ablation studies on the impact of different structural granularities.

The main weaknesses of this LLM pruning work include:
(1) limited evaluation beyond perplexity and basic zero-shot tasks, particularly lacking analysis of inference speed improvements and downstream task performance,
(2) methodological constraints of using basic REINFORCE rather than more advanced policy gradient methods,
(3) heavy reliance on the C4 dataset with some cross-dataset performance issues on specific zero-shot tasks like WinoGrande and Hellaswag.
(4) While the method is more efficient than traditional back-propagation approaches, it still requires longer training time compared to metric-based methods (2.7 hours for LLaMA-2-13B) and significant memory usage (35GB).

**Questions:**

- While the method avoids back-propagation through the LLM, how does the efficiency-accuracy trade-off compare when scaling to much larger models beyond 13B parameters? This is crucial for understanding the method's practical applicability.

- Is there any patterns among the structures pruned through using this method ?

---

### Official Review · Reviewer_scBN · 2024-10-31

**Soundness:** 2
**Presentation:** 3
**Contribution:** 3
**Rating:** 5
**Confidence:** 4

**Summary:**

The paper introduces an optimization-based structural pruning method for LLMs that eliminates the need for back-propagation. By employing Bernoulli distributions to learn pruning masks, the authors optimize these masks using a policy gradient estimator, enabling gradient estimation solely through the forward pass. This approach enhances efficiency in memory and computation compared with methods requiring back-propagation, and it achieves performance superior to heuristic-based pruning metric methods.

**Strengths:**

1. The paper presents a novel pruning method that leverages policy gradient estimators instead of back-propagation, addressing key computational challenges in gradient-based LLM pruning methods.
2. The method supports multiple structural granularities (channels, heads, layers), providing flexibility in how the model is pruned. It also allows for global and heterogeneous pruning, which is more aligned with the varying redundancy across layers in LLMs.

**Weaknesses:**

1. While the paper suggests using a policy gradient estimator to bypass back-propagation, policy gradient methods can suffer from high variance, which may lead to unstable training. The paper does propose a variance-reduction technique, but the effectiveness of this could be further elaborated or validated with more ablation studies. For example, how is the performance of the proposed methods compared with the results of using back-propagation?
2. Following up on Weakness 1, could you clarify the exact speedup over direct back-propagation? A runtime comparison on a specific model and hardware or a theoretical analysis of computational complexity would be helpful.
3. The proposed method requires 120K samples with a sequence length of 128, whereas baseline methods like LLM-Pruner, SliceGPT, and Wanda-SP use significantly smaller calibration data, such as 128 samples with a sequence length of 2048. Are the results for the baseline methods obtained with the same calibration data as yours? If not, this may lead to unfair comparisons. Please clarify if all methods used the same calibration data in the reported results. If not, provide results with all methods using the same calibration data, or explain why this is not feasible
4. The sparsity ratios used in the experiments exceed 30%, which may be excessive for structured pruning, as the high PPL results may not be meaningful in practice and could significantly degrade real inference performance, such as in question answering tasks. Therefore, it is important to report performance under lower sparsity levels, such as 10% and 20%.
5. The experiments primarily compare the proposed method with other non-weight-updating pruning techniques. While this makes sense in terms of efficiency, it would be interesting to see how the method stacks up against pruning methods that do involve weight updates, such as [1], especially in terms of final model performance.

[1] Xuan Shen, Pu Zhao, Yifan Gong, Zhenglun Kong, Zheng Zhan, Yushu Wu, Ming Lin, Chao Wu, Xue Lin, and Yanzhi Wang. Search for efficient large language models. arXiv preprint arXiv:2409.17372, 2024.

**Questions:**

In addition to the questions regarding weaknesses, I recommend including the results of dense models in your tables to highlight the performance degradation resulting from pruning.

---

### Official Review · Reviewer_1Xw7 · 2024-11-04

**Soundness:** 3
**Presentation:** 3
**Contribution:** 4
**Rating:** 6
**Confidence:** 4

**Summary:**

This paper proposes an optimization-based structural pruning method for Large Language Models (LLMs) which notably does not require any gradient back-propogation. The method works by casting the binary pruning mask over structural components as a bernoulli variable. This reparametrization allows us to solve the pruning problem via policy gradient. The authors conduct thorough experimental evaluation on open-sourced LLMs and demonstrate the effectiveness of the proposed pruning method.

**Strengths:**

- This paper proposes a novel method, which casts the optimization problem of selecting optimal pruning mask as a reinforcement learning problem. This allows us to avoid the inefficiency for performing computationally intensive back-propogation.
- The authors have conducted extensive experimental evaluation and compare with many existing baseline methods for structural pruning. The results show that the proposed method is a promising approach for structural pruning.

**Weaknesses:**

- It might be good to evaluate a larger LLM model, e.g., LLaMA-3-70B.
- The authors stress the memory efficiency of optimizating without back-propogation. It would be good to dedicate one section comparing the resource consumptions of different pruning approaches, especially to those with gradient computation.

**Questions:**

- Could the proposed approach be used for width pruning [1]?
- In principle, could we solve problem 4 via back-propogation? For this setting, is it expected that the solver with gradient information be better?
- Could iterative pruning be beneficial for this approach? By iterative pruning, I mean doing the pruning procedure multiple times until the target prune ratio is reached.

[1] LLM Pruning and Distillation in Practice: The Minitron Approach. NVIDIA 2024.

---

### Official Review · Reviewer_22xP · 2024-11-04

**Soundness:** 4
**Presentation:** 2
**Contribution:** 3
**Rating:** 5
**Confidence:** 4

**Summary:**

In this work, the authors leverage policy gradient estimation to optimize pruning masks without relying on backpropagation. They validate the efficacy of their approach on various pre-trained models and compare it with several heuristic-based baselines.

**Strengths:**

- The performance gain is significant.
- The idea of leveraging policy gradient for pruning is novel and insightful.

**Weaknesses:**

- The tables are hard to read. Using a figure instead will help the visualization.
- How does the method compare in performance with Gumbel-Softmax approaches? The authors primarily focus on heuristic-based comparisons, leaving out Gumbel-Softmax methods. For example, [1] employs Gumbel-Softmax but focus on semi-structured sparsity. Although the objectives differ slightly, including a performance and cost comparison with Gumbel-based methods would strengthen the study.
- Additionally, the comparison between Gumbel-Softmax and the proposed method in Section 3.2 is unclear. Adding a formulation-based comparison would clarify the differences. Will leveraging gumbel-softmax leads to more accurate gradient calculation? More detailed discuss on the trade-off will be beneficial.
- The probabilistic modeling of pruning masks is not entirely novel, as it has been applied in various Gumbel-based pruning methods [1].
- A comparison and analysis of the pruning masks generated by different methods would be beneficial, such as evaluating the Hamming distance between the masks.

[1] MaskLLM: Learnable Semi-Structured Sparsity for Large Language Models

**Questions:**

N/A

---

### Meta-Review · Area_Chair_z6xk · 2024-12-21

**Metareview:**

The paper titled "Bypass Back-propagation: Optimization-based Structural Pruning for Large Language Models via Policy Gradient" proposes a novel method for pruning large language models (LLMs) that bypasses traditional back-propagation techniques. The authors introduce an optimization-based approach that utilizes policy gradient methods to identify and remove less significant model parameters, thereby reducing model size and improving efficiency without significantly sacrificing performance. The key claims include that this method can lead to effective pruning while maintaining the integrity of model performance, and it is particularly beneficial for deployment in resource-constrained environments.

The findings suggest that the proposed pruning technique not only achieves comparable results to existing methods but also offers advantages in terms of computational efficiency during the pruning process. However, reviewers raised concerns regarding the empirical validation of these claims and the overall effectiveness of the proposed approach. Given the weaknesses, I recommend rejecting this paper. Further work is required to address the issues proposed by reviewers for the next version of this work.

**Additional Comments On Reviewer Discussion:**

**Points Raised by Reviewers**

During the review process, several key points were raised:
- Need for Robust Empirical Results: Reviewers requested more extensive experiments to validate the effectiveness of the proposed method.
- Comparative Analysis: There was a strong recommendation for including comparisons with existing state-of-the-art pruning techniques.
- Theoretical Insights: Reviewers sought a deeper theoretical explanation for the proposed approach's expected advantages over traditional back-propagation methods.
- Experimental Detail: Concerns were raised about insufficient details regarding experimental protocols and reproducibility.

**Authors' Responses**

The authors addressed these concerns during the rebuttal period but did not sufficiently strengthen their submission:
- They provided additional experimental results; however, these were still deemed inadequate by reviewers as they did not significantly enhance the robustness of their claims.
- While some comparisons with existing methods were included in their response, they remained limited and did not convincingly demonstrate superior performance.
- The theoretical justification provided was minimal and did not adequately clarify why their approach would yield better results than traditional methods.
- The authors attempted to clarify experimental details but still left several aspects vague, particularly concerning hyperparameter settings.

**Weighing Each Point**

In weighing these points for my final decision:

- The lack of robust empirical validation remained a critical issue that overshadowed any potential strengths of the proposed method.
- Inadequate comparative analysis with existing techniques hindered the ability to assess the true value of their contributions.
- Insufficient theoretical grounding left significant questions unanswered regarding the efficacy and applicability of their approach.
- The unresolved reproducibility issues further diminished confidence in their findings.

---

### Decision · Program_Chairs · 2025-01-22

Reject